# CYLD-TRAF6 interaction promotes ADP-heptose-induced NF-κB signaling in *H. pylori* infection

Michelle C C Lim, Gunter Maubach ⓘD & Michael Naumann ⓘD ✉

## Abstract

The inflammatory response associated with *Helicobacter pylori* (*H. pylori*) infection causes a multitude of alterations in the gastric microenvironment, leading to the slow and steady disruption of the gastric epithelial barrier. Activation of NF-κB during *H. pylori* infection is crucial to this inflammatory response. Here, we show that CYLD, which interacts constitutively with TRAF6, enhances *H. pylori's* ADP-heptose-induced activation of the classical NF-κB pathway in gastric epithelial cells. This activating effect of CYLD contrasts with the inhibitory effect of CYLD on receptor-mediated NF-κB activity. Mechanistically, CYLD counteracts the hydrolysis of ubiquitin chains from TRAF6 by deubiquitinylase A20 in a catalytically independent manner, thus supporting the auto-ubiquitinylation of TRAF6 upon activation of NF-κB in early *H. pylori* infection. In addition, the subsequent classical NF-κB-dependent de novo synthesis of A20 provides a negative feedback loop leading to shutdown not only of the classical but also of the alternative NF-κB pathway. Our findings highlight the regulatory relationship between CYLD and A20 in controlling classical as well as alternative NF-κB signaling in *H. pylori* infection.

**Keywords** ALPK1; A20; Gastric Spheroid; TIFA
**Subject Categories** Microbiology, Virology & Host Pathogen Interaction; Post-translational Modifications & Proteolysis; Signal Transduction

## Introduction

*Helicobacter pylori* is one of the most successful human pathogens with a worldwide prevalence of 43% (Li et al, 2023). The only known niche of this Gram-negative bacterium is the human stomach, where it colonizes the gastric mucosal epithelium. Unless eradicated, infection with *H. pylori* is persistent and is associated with chronic inflammation, leading to chronic active gastritis. In addition, infection with *H. pylori* represents a major risk factor for the development of gastric malignancies (Park et al, 2018).

Multiple bacterial factors have been implicated in the induction of NF-κB upon *H. pylori* infection (Backert and Naumann, 2010). Recently, adenosine diphosphate (ADP)-l-glycero-β-d-manno-heptose (ADP-heptose) has emerged as an immunomodulator of

Gram-negative bacteria and is defined as a pathogen-associated molecular pattern (Sidor and Skirecki, 2023). ADP-heptose, an intermediate metabolite of the lipopolysaccharide biosynthetic pathway, is produced in a four-step process catalyzed by nucleotide diphosphate-heptose biosynthetic enzymes that are distributed widely in bacteria, archaea, eukaryotes, and viruses (Tang et al, 2024). During *H. pylori* infection, ADP-heptose enters the cell to activate nuclear factor kappa-light-chain-enhancer of activated B cells (NF-κB) (Pfannkuch et al, 2019), a critical regulator of the pro-inflammatory responses underlying the pathogenicity of *H. pylori* (Maubach et al, 2022). ADP-heptose binds to alpha-protein kinase 1 (ALPK1) in the cytosol. This interaction activates ALPK1 which phosphorylates the threonine residue at position 9 (T9) of tumor necrosis factor receptor-associated factor (TRAF)-interacting protein with forkhead-associated domain (TIFA) dimers (Snelling et al, 2022), inducing the formation of so-called TIFAsomes that were postulated to be TIFA oligomers (Zimmermann et al, 2017) or TIFA undergoing dynamic liquid–liquid phase separation (Li et al, 2024). In *H. pylori* infection, the subsequent interaction of TIFA with TRAF6 or TRAF2 molecules activate the classical and the alternative NF-κB pathways, respectively (Maubach et al, 2021). Binding of TIFA to TRAF6 recruits and activates the transforming growth factor (TGF)-β-activated kinase 1 (TAK1) in complex with TAK1-binding proteins (TABs). This leads to activation of the inhibitor of nuclear factor κB kinase (IKK) complex and therefore the classical NF-κB pathway. Binding of TIFA with TRAF2 in the NF-κB-inducing kinase (NIK) regulatory complex (comprising TRAF3, TRAF2, and cellular inhibitor of apoptosis protein 1 (cIAP1) or cIAP2), on the other hand, promotes cIAP1 degradation. This leads to the stabilization of NIK to initiate activation of the alternative NF-κB pathway.

The cylindromatosis (CYLD) protein is a deubiquitinylase (DUB) belonging to the ubiquitin-specific protease superfamily that cleaves preferentially K63- and M1-linked polyubiquitin (Ub) chains (Komander et al, 2008; Ritorto et al, 2014; Sato et al, 2015). CYLD acts as a negative regulator of cytokine-stimulated NF-κB signaling as several NF-κB signaling components have been proposed to be its substrates (Kovalenko et al, 2003; Trompouki et al, 2003; Yoshida et al, 2005). Consistently, not only did lymphocytes and macrophages from *Cyld*⁻/⁻ mice exhibit elevated NF-κB activation in response to mediators of innate and adaptive immunity, but these mice also showed exacerbated inflammation and subsequent tumor formation in a colitis-associated cancer model (Zhang et al, 2006). The DUB activity of CYLD towards K63-Ub is regulated through phosphorylation. For instance, a

Otto von Guericke University, Institute of Experimental Internal Medicine, Medical Faculty, Leipziger Str. 44, 39120 Magdeburg, Germany. ✉E-mail: Naumann@med.ovgu.de

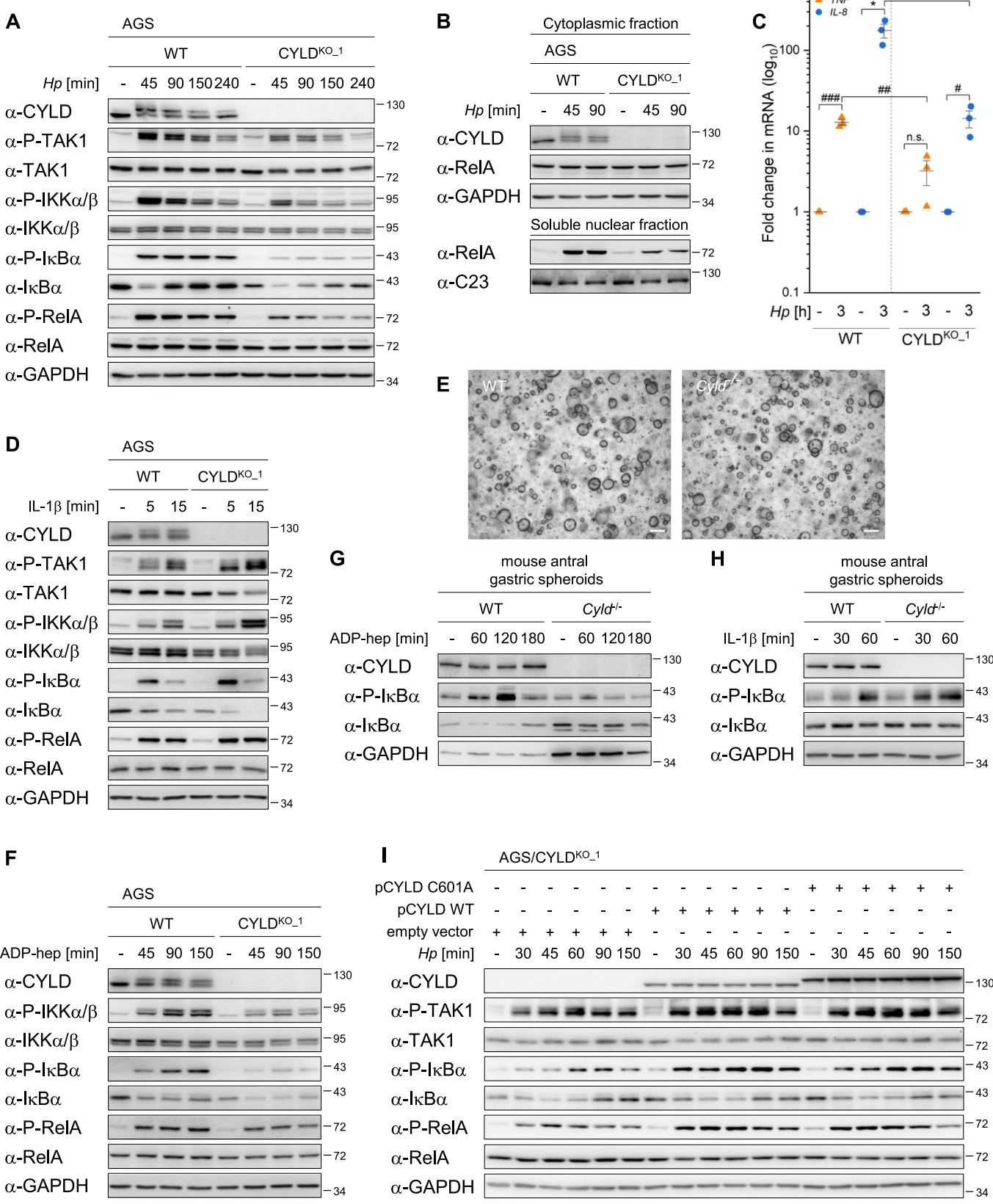

**Figure 1.** CYLD supports *H. pylori*-induced classical NF-κB activation.

(A) WT and CYLD$^{KO\_1}$ AGS cells were left uninfected (−) or infected with *H. pylori* for the times shown. Total lysates were analyzed by immunoblotting using the indicated antibodies. (B) WT and CYLD$^{KO\_1}$ AGS cells were left uninfected or infected with *H. pylori* for the times shown. Cell fractionation was performed to obtain the cytoplasmic and soluble nuclear fractions before analysis by immunoblotting using the indicated antibodies. GAPDH and C23 serve as the loading controls for the cytoplasmic and soluble nuclear fractions, respectively. (C) WT and CYLD$^{KO\_1}$ AGS cells were left uninfected or infected with *H. pylori* for 3 h. Total RNA was extracted and reverse transcribed to cDNA. Changes in *TNF* and *IL-8* transcripts were investigated by quantitative PCR and expressed as fold change relative to the respective uninfected sample (set to 1). The data and error bars represent means and SDs from three biological replicates, respectively. $^{*}P < 0.01$, $^{#}P < 0.05$, $^{##}P < 0.005$, $^{###}P < 0.0005$, n.s. = not significant (unpaired two-tailed Student's *t* test). (D) WT and CYLD$^{KO\_1}$ AGS cells were left untreated or treated with 10 ng/ml IL-1β for the indicated times. Total lysates were analyzed by immunoblotting using the indicated antibodies. (E) Representative bright-field images of spheroids established from isolated antral stomach glands of WT and *Cyld*$^{-/-}$ mice. Scale bars: 200 μm. (F) WT and CYLD$^{KO\_1}$ AGS cells were left untreated or treated with 200 nM ADP-heptose for the indicated times. Total lysates were analyzed by immunoblotting using the indicated antibodies. (G) Antral gastric spheroids of WT and *Cyld*$^{-/-}$ mice were left untreated or treated with 200 nM ADP-heptose for the times shown. Total lysates were analyzed by immunoblotting using the indicated antibodies. (H) Antral gastric spheroids of WT and *Cyld*$^{-/-}$ mice were left untreated or treated with 10 ng/ml IL-1β for the times shown. Total lysates were analyzed by immunoblotting using the indicated antibodies. (I) CYLD$^{KO\_1}$ AGS cells were transfected with empty vector, plasmids expressing WT CYLD (pCYLD WT), or catalytically inactive CYLD mutant (pCYLD C601A) 24 h prior to infection with *H. pylori* for the times shown. Overexpressed WT CYLD and C601A CYLD have different electrophoretic mobilities because they carry different N-terminal tags (see "Methods"). For (A, D, F–I), GAPDH serves as the loading control for the total lysates. Immunoblots (IBs) were processed in parallel and depict 1 representative of at least two independent experiments. Source data are available online for this figure.

recent report showed that phosphorylation at both S418 and S568 following TNF stimulation activated CYLD's K63-Ub-directed DUB activity (Elliott et al, 2021). This activity is also regulated through phosphorylation of multiple serine residues in a serine cluster (amino acids 411–444), although it is controversial as to whether this has a promoting or suppressing effect (Hutti et al, 2009; Reiley et al, 2005; Thein et al, 2014).

A20, encoded by the NF-κB-inducible tumor necrosis factor alpha-induced protein 3 (*TNFAIP3*) gene, is a DUB that inhibits NF-κB signaling in a negative feedback mechanism. A20 contains an N-terminal ovarian tumor domain that exhibits strong activity towards K63-ubiquitinylated NF-κB signaling intermediates in cells (Boone et al, 2004; Hitotsumatsu et al, 2008; Wertz et al, 2004). In addition, non-catalytic mechanisms of action of A20 have been described, including stabilization of M1-Ub chains through its zinc finger 7 domain and the disruption of E3 ubiquitin ligase/E2 enzyme complexes to limit polyubiquitin chain formation (Martens et al, 2020; Shembade et al, 2010; Tokunaga et al, 2012). Interestingly, in *H. pylori* infection, A20 participates in the regulation of both NF-κB and cell survival (Jantaree et al, 2022; Lim et al, 2022; Lim et al, 2017), and this has likely implication for the infection persistence.

The involvement of CYLD in the regulation of ubiquitin chains in NF-κB signaling led us to question whether it plays a role during *H. pylori* infection. Surprisingly, we found that, contrary to its inhibitory role in cytokine-induced classical NF-κB pathway, CYLD participates in the activation of the classical NF-κB pathway during *H. pylori* infection. Subsequently, the resulting upregulation of A20 contributes to the shutdown of *H. pylori*-induced alternative NF-κB pathway.

## Results and discussion

### CYLD supports *H. pylori*-induced classical NF-κB signaling

CYLD has been shown to negatively regulate NF-κB activation by different inducers, including TNF, IL-1β, and CD40 (Jono et al, 2004; Kovalenko et al, 2003; Trompouki et al, 2003). To investigate the involvement of CYLD in *H. pylori*-induced classical NF-κB activation, we used CRISPR/Cas9 gene-editing technology to

generate CYLD-deficient AGS cells, henceforth termed CYLD-knockout (CYLD$^{KO}$) AGS cells. AGS cells subjected to the same CRISPR/Cas9 procedure but without successful *CYLD* knockout are referred to as wild-type (WT) cells. We were surprised by the observation that activation of classical NF-κB infection upon infection with *H. pylori* P1 strain was prominently impaired in CYLD$^{KO}$ cells, as monitored by the (activation) phosphorylation of TAK1 and IKKα/β, their downstream targets IκBα and RelA (Fig. 1A) as well as the nuclear translocation of RelA (Fig. 1B). Accordingly, the induction of NF-κB-regulated *TNF* and *IL-8* transcripts was attenuated in infected CYLD$^{KO}$ cells (Fig. 1C). Comparable results were obtained with a second clone of CYLD$^{KO}$ cells (Fig. EV1A,B) and CYLD-depleted AGS cells by transfection using two different CYLD-specific siRNAs separately (Fig. EV1C,D), ruling out off-target effects due to the methods used for the ablation of CYLD. Diminished classical NF-κB activation in infected CYLD$^{KO}$ cells was also observed with another strain of *H. pylori* (Fig. EV1E). In addition, we noted significantly reduced classical NF-κB activation in *H. pylori*-infected CYLD-deficient NCI-N87 cells (Fig. EV1F), thus excluding cell line-specific effects. Notably, CYLD was detected as double bands upon *H. pylori* infection, where the slower migrating band corresponds to phosphorylated CYLD as evidenced by its disappearance after λ phosphatase treatment (Fig. EV1G). This observation is in accordance with the reported phosphorylation(s) of CYLD within a serine cluster (amino acid 392–444) during NF-κB activation in response to cellular stimuli (Elliott et al, 2021; Reiley et al, 2005). Consistent with the reported inhibitory role of CYLD towards NF-κB activation induced by members of the IL-1 receptor/Toll-like receptor families (Kovalenko et al, 2003), we observed enhanced classical NF-κB activation in IL-1β-treated CYLD$^{KO}$ AGS cells (Figs. 1D and EV1H). So far, our data indicated an unusual role for CYLD in the activation of the classical NF-κB pathway by *H. pylori* infection.

Next, we examined the role of CYLD in a setting close to in vivo by utilizing gastric epithelial spheroids that were established from isolated antral stomach glands of WT and *Cyld*$^{-/-}$ mice. The spheroids from both mice appeared similar (Fig. 1E); however, we noticed that in general, those from *Cyld*$^{-/-}$ mice grew faster. One explanation could be the function of CYLD as a negative regulator of cell cycle progression (Wickström et al, 2010). The apical

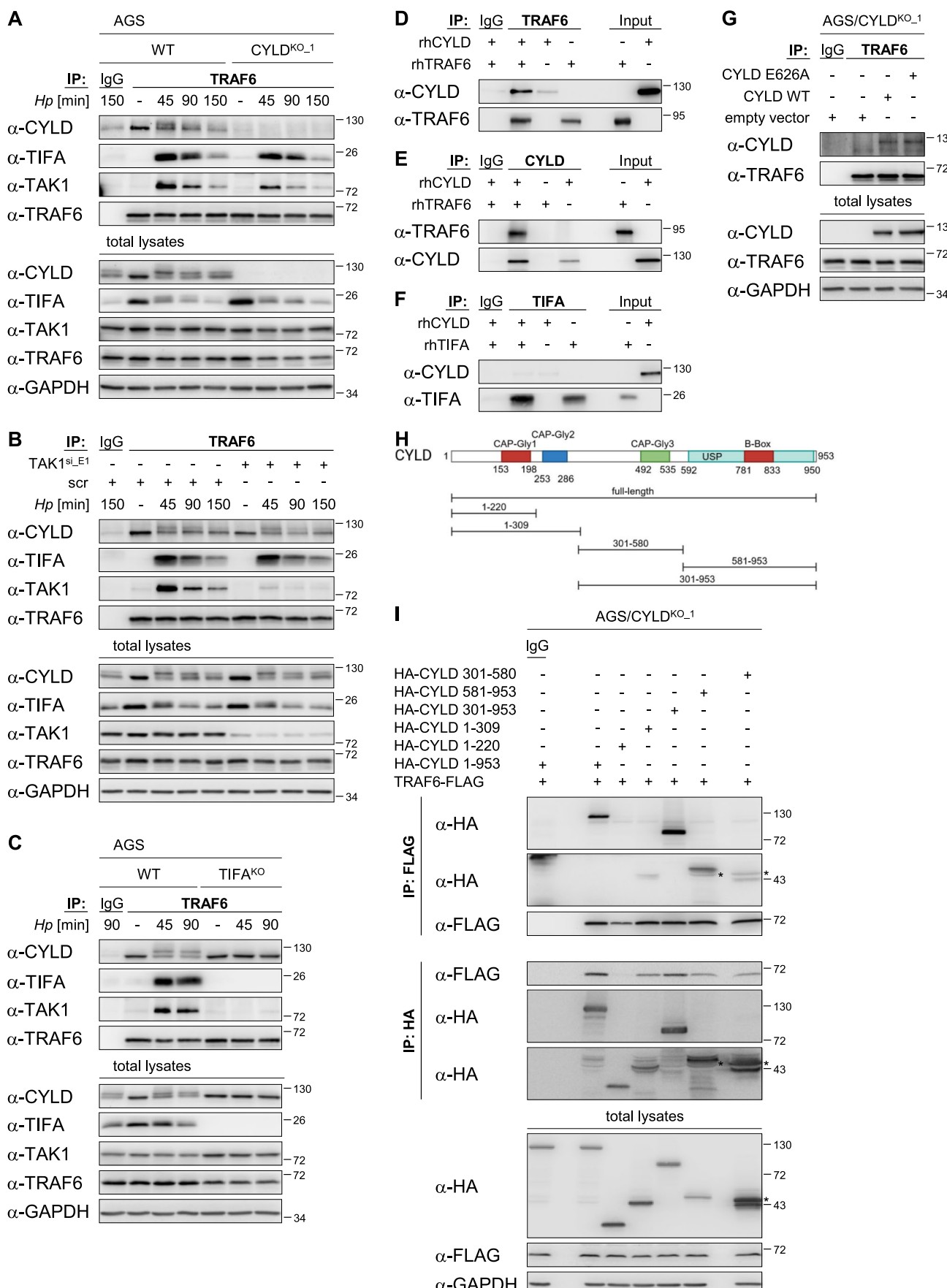

◄  **Figure 2. CYLD associates with the TIFA/TRAF6/TAK1 complex upon *H. pylori* infection.**

(A) WT and CYLD^KO_1 AGS cells were left uninfected (-) or infected with *H. pylori* for the times shown and harvested for total lysates. IP with an anti-TRAF6 antibody or an isotype-matched antibody (IgG) was performed. Eluates and total lysates were analyzed by immunoblotting using the indicated antibodies. (B) AGS cells were transfected with siRNAs targeting TAK1 (TAK1^si_E1, 20 nM) or a non-targeting scrambled siRNA control (scr, 20 nM) 48 h prior to infection with *H. pylori* for the times shown. Total lysates were subjected to IP with an anti-TRAF6 antibody or an isotype-matched antibody (IgG). Eluates and total lysates were analyzed by immunoblotting using the indicated antibodies. (C) WT and TIFA^KO AGS cells were left uninfected or infected with *H. pylori* for the times shown. Total lysates were subjected to IP with an anti-TRAF6 antibody or an isotype-matched antibody (IgG). Eluates and total lysates were analyzed by immunoblotting using the indicated antibodies. (D) Following incubation of recombinant human (rh) CYLD and rhTRAF6 proteins in vitro, IP was performed using an antibody against TRAF6. Eluates and input (10 ng rhCYLD or rhTRAF6 proteins) were analyzed by immunoblotting using the indicated antibodies. (E) Following incubation of rhCYLD and rhTRAF6 proteins in vitro, IP was performed using an antibody against CYLD. Eluates and input (10 ng rhCYLD or rhTRAF6 proteins) were analyzed by immunoblotting using the indicated antibodies. (F) Following incubation of rhCYLD and rhTIFA proteins in vitro, IP was performed using an antibody against TIFA. Eluates and input (10 ng rhCYLD or rhTIFA proteins) were analyzed by immunoblotting using the indicated antibodies. (G) CYLD^KO_1 AGS cells were transfected with empty vector, plasmids expressing WT CYLD (CYLD WT), or CYLD mutant with a disrupted TRAF6-binding motif (CYLD E626A) 48 h prior to total lysates preparation. IP was performed with an anti-TRAF6 antibody or an isotype-matched antibody (IgG). Eluates and total lysates were analyzed by immunoblotting using the indicated antibodies. (H) Schematic representation of the CYLD constructs generated for overexpression in CYLD^KO_1 cells. (I) CYLD^KO_1 AGS cells were transfected with plasmids expressing HA-tagged full-length CYLD (1-953) or the different CYLD variants together with FLAG-tagged TRAF6 for 24 h prior to total lysate preparation. Total lysates were subjected to IP with an anti-FLAG antibody, anti-HA antibody or isotype-matched antibody (IgG). Eluates and total lysates were analyzed by immunoblotting using the indicated antibodies. Asterisk denotes an unspecific band. For (A–C, G, I), GAPDH serves as the loading control for the total lysates. IBs were processed in parallel and depict 1 representative of at least two independent experiments. Source data are available online for this figure.

epithelium surface, where infection by *H. pylori* typically occurs (Boccellato et al, 2019), is inaccessible to the bacteria because it is on the luminal side of the spheroids. Therefore, we treated these gastric spheroids with ADP-heptose that was previously shown to induce NF-κB in the context of *H. pylori* infection (Pfannkuch et al, 2019). Consistent with our observations using *H. pylori* infection (Fig. 1A), ADP-heptose-triggered NF-κB activation was attenuated in CYLD^KO cells compared to WT cells (Fig. 1F). CYLD-deficient spheroids exhibited markedly abated classical NF-κB activation by ADP-heptose treatment (Figs. 1G and EV1I), while the opposite was observed for IL-1β stimulation (Fig. 1H), which is in line with our data from cell line studies.

CYLD has been implicated as a negative regulator of TNF-induced NF-κB activation due to its ability to hydrolyze both M1- and K63-Ub from components of the TNF receptor proximal signaling complex (Draber et al, 2015; Kovalenko et al, 2003). M1-Ub are generated by the linear ubiquitin chain assembly complex (LUBAC) (Ikeda et al, 2011; Kirisako et al, 2006; Tokunaga et al, 2011). Knockdown of HOIP, the catalytic component of LUBAC (Stieglitz et al, 2013), led to a marginal reduction in the phosphorylation of IKKα/β, IκBα, and RelA (Fig. EV1J), suggesting that M1-Ub has a minor contribution to *H. pylori*-induced classical NF-κB activation. The K63-linked ubiquitinylation of TRAF6 is required for propagating the signal for activating classical NF-κB in a TIFA-dependent manner (Ea et al, 2004). Ubiquitinylated TRAF6 was proposed to be a target of CYLD's deubiquitinylase activity (Trompouki et al, 2003). Overexpression of WT CYLD as well as catalytically inactive CYLD (C601A) in CYLD^KO cells enhanced classical NF-κB signaling by *H. pylori* infection (Fig. 1I), indicating that CYLD's role in this pathway is not strictly dependent on its deubiquitinylase activity. While this does not entirely preclude a role for CYLD in modulating M1- and K63-linked ubiquitin chains during *H. pylori* infection, such activity appears to be of secondary relevance compared to its function in regulating receptor-mediated NF-κB activation. These findings suggest that CYLD may influence classical NF-κB signaling through mechanisms beyond direct ubiquitin editing, potentially implicating scaffolding or regulatory interactions that remain to be fully elucidated.

## CYLD associates with the TIFA/TRAF6/TAK1 complex upon *H. pylori* infection

We have reported previously that the TIFA/TRAF6/TAK1 complex is necessary for recruitment and activation of the IKK complex to induce NF-κB in *H. pylori* infection (Maubach et al, 2021; Sokolova et al, 2014). Therefore, our observations regarding the effect of CYLD deficiency early in the *H. pylori*-induced NF-κB signaling cascade, i.e., already on the level of TAK1 phosphorylation, sparked our interest to examine CYLD protein–protein interactions at the TIFA/TRAF6/TAK1 complex. Using a TRAF6-directed antibody for immunoprecipitation (IP), we observed in WT cells the constitutive interaction of endogenous CYLD with TRAF6 that decreased moderately following *H. pylori* infection (Fig. 2A). Upon *H. pylori* infection, phosphorylated CYLD also appeared in the complex (Fig. 2A). Further analysis of TRAF6-immunoprecipitates revealed inducible association with TIFA and TAK1 in *H. pylori*-infected WT cells, in agreement with our earlier findings (Maubach et al, 2021; Sokolova et al, 2014), and, intriguingly, this was reduced in *H. pylori*-infected CYLD^KO cells (Figs. 2A and EV2A). Similar effects were observed using ADP-heptose as a trigger (Fig. EV2B). Reciprocal IP with a TIFA-directed antibody confirmed inducible association with TRAF6 and CYLD (Fig. EV2C). Reducing the abundance of TAK1 using a RNA interference approach affected neither the interaction between CYLD and TRAF6 nor, expectedly, between TIFA and TRAF6 (Fig. 2B). Using TIFA-knockout (TIFA^KO) cells (Maubach et al, 2021), we observed that TIFA was not required for the interaction between CYLD and TRAF6 (Fig. 2C). In contrast, TAK1 interacted with TRAF6 upon *H. pylori* infection only in the presence of TIFA. We also observed that CYLD phosphorylation was absent in TIFA^KO cells (Fig. 2C). According to Elliot and colleagues (Elliott et al, 2021), CYLD phosphorylation is dependent on NF-κB activation, which is abolished in TIFA^KO cells due to the missing interaction of TIFA with TRAF6 and TAK1. Collectively, our results strongly indicated the constitutive interaction of CYLD with TRAF6 in the TIFA/TRAF6/TAK1 complex in *H. pylori* infection.

When we co-incubated recombinant human CYLD and TRAF6 proteins in vitro, we could detect CYLD in the TRAF6-

immunoprecipitates (Fig. 2D) and vice versa (Fig. 2E). Using recombinant human CYLD and TIFA proteins in the same setup, no CYLD was detected in TIFA-immunoprecipitates (Fig. 2F), erasing doubts of artifactual interaction due to such an experimental design. These results showed explicitly a direct interaction between CYLD and TRAF6. CYLD has a potential TRAF6-binding

motif xxxPxExx[FYWHDE] (Halpin et al, 2022) consisting of the core sequence LLRPKEKND that might be necessary for binding to TRAF6 (Darnay et al, 1999; Pullen et al, 1999). We figured that by re-introducing a CYLD E626A mutant (the site of mutation was chosen by extrapolation from the TIFA E178A mutant that has a disrupted TRAF6-binding motif SSSPTEMDE (Takatsuna et al,

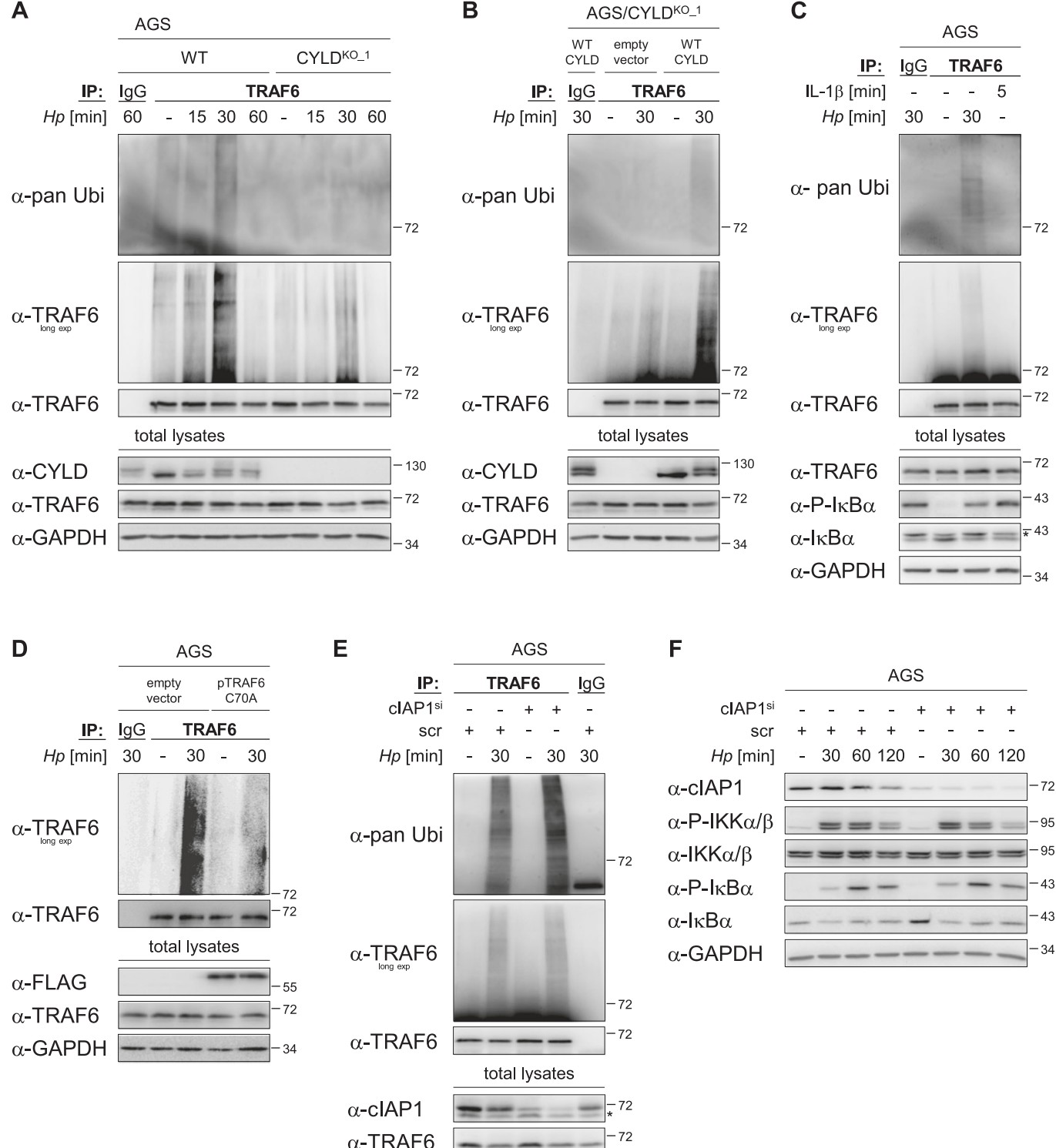

◀ **Figure 3.  CYLD stabilizes TRAF6 ubiquitinylation.**

(A) WT and CYLD^(KO-1) AGS cells were left uninfected (-) or infected with *H. pylori* for the times shown and harvested for total lysates using lysis buffer containing 1% SDS (denaturing condition). IP with an anti-TRAF6 antibody or an isotype-matched antibody (IgG) was performed. Eluates and total lysates were analyzed by immunoblotting using the indicated antibodies. (B) Stable clones of CYLD^(KO-1) AGS cells with pSelect empty vector or WT CYLD were left uninfected or infected with *H. pylori* for 30 min and harvested for total lysates using lysis buffer containing 1% SDS (denaturing condition). IP was performed using an anti-TRAF6 antibody or an isotype-matched antibody (IgG). Eluates and total lysates were analyzed by immunoblotting using the indicated antibodies. (C) AGS cells were infected with *H. pylori* for 30 min or stimulated with 10 ng/ml IL-1β for 5 min and harvested for total lysates using lysis buffer containing 1% SDS (denaturing condition). IP was performed using an anti-TRAF6 antibody or an isotype-matched antibody (IgG). Eluates and total lysates were analyzed by immunoblotting using the indicated antibodies. (D) CYLD^(KO-1) AGS cells were transfected with empty vector or plasmid expressing the catalytically inactive TRAF6 mutant (pTRAF6 C70A) 48 h prior to infection with *H. pylori* for 30 min and total lysates were harvested using lysis buffer containing 1% SDS (denaturing condition). IP was performed using an anti-TRAF6 antibody or an isotype-matched antibody (IgG). Eluates and total lysates were analyzed by immunoblotting using the indicated antibodies. (E) AGS cells were transfected with siRNAs targeting cIAP1 (cIAP1^si, 25 nM) or a non-targeting scrambled siRNA control (scr, 25 nM) 48 h prior to infection with *H. pylori* for 30 min and harvested for total lysates using lysis buffer containing 1% SDS (denaturing condition). IP with an anti-TRAF6 antibody or an isotype-matched antibody (IgG) was performed. Eluates and total lysates were analyzed by immunoblotting using the indicated antibodies. (F) AGS cells were transfected with siRNAs targeting cIAP1 (cIAP1^si, 25 nM) or a non-targeting scrambled siRNA control (scr, 25 nM) 48 h prior to infection with *H. pylori* for the times shown. Total lysates were analyzed by immunoblotting using the indicated antibodies. For all panels, GAPDH serves as the loading control for the total lysates. IBs were processed in parallel and depict 1 representative of at least two independent experiments. Asterisk denotes an unspecific band. Source data are available online for this figure.

2003)) into CYLD^(KO) cells, it would be possible to investigate whether CYLD binds to TRAF6 via its TRAF6-binding motif. CYLD WT as well as CYLD E626A mutant co-immunoprecipitated with TRAF6 (Fig. 2G), suggesting that the interaction between CYLD and TRAF6 does not require CYLD's TRAF6-binding motif. This observation is in line with data indicating that TIFA interacts with TRAF6 via TIFA's TRAF6-binding motif (Takatsuna et al, 2003), hence excluding CYLD from binding at the same site. The CYLD's TRAF6-binding sequence contains a positively charged lysine residue at position +3 that destabilizes any interaction via this sequence (Halpin et al, 2022). As a result, TIFA would be in favor for the binding to TRAF6. The detected possible ubiquitinylation of K627 adjacent to E626 (Akimov et al, 2018) might be another reason that prevents the binding of CYLD via its TRAF6-binding domain.

To further elucidate the region of CYLD that is implicated in the interaction with TRAF6, we transfected plasmids expressing HA-tagged full-length and truncated variants of human CYLD into CYLD^(KO) cells together with FLAG-tagged TRAF6 (Fig. 2H,I) and analyzed their binding ability to TRAF6 by performing a FLAG or HA IP. We precipitated the full-length as well as truncated variants of CYLD (Fig. 2I) with the exception of the variant containing only the first CAP-Gly domain (1–220). This finding suggested that the N-terminal portion of CYLD is not required for its interaction with TRAF6.

## CYLD counteracts A20-dependent deubiquitinylation of TRAF6

The interaction of TIFA and TRAF6 is essential in classical NF-κB activation in *H. pylori* (Maubach et al, 2021) as well as *Shigella flexneri* infections, which involved the formation of TIFA/TRAF6 oligomers upon phosphorylation at T9 of TIFA (Milivojevic et al, 2017). The promotion of TRAF6 oligomerization by TIFA and the resulting increase in TRAF6's E3 ligase activity was clearly demonstrated (Ea et al, 2004). Furthermore, oligomerization of TRAF6 facilitates interaction with the ubiquitin-conjugating enzyme (E2) Ubc13 leading to TRAF6 auto-ubiquitinylation (Lamothe et al, 2007; Yin et al, 2009). Incubation of the cells with a selective TRAF6-Ubc13 inhibitor, C25-140 (Brenke et al, 2018), prior to *H. pylori* infection led to the reduced phosphorylation of TAK1 and IKKα/β (Fig. EV3A), implicating TRAF6 auto-

ubiquitinylation in *H. pylori*-induced classical NF-κB activation. Given that ADP-heptose treatment also resulted in TRAF6 ubiquitinylation (Snelling et al, 2022), we examined the role of CYLD in this ubiquitinylation event. As expected, inducible and transient TRAF6 ubiquitinylation was detected in *H. pylori*-infected WT cells (Fig. 3A). Interestingly, TRAF6 ubiquitinylation was considerably diminished in *H. pylori*-infected CYLD^(KO) cells (Fig. 3A) and this was restored by the re-expression of WT CYLD as well as catalytically inactive CYLD (Figs. 3B and EV3B). Thus far, we see a correlation between CYLD expression and the extent of TRAF6 ubiquitinylation in *H. pylori* infection. TRAF6, which is essential in the IL-1R signaling pathway, was not ubiquitinylated after IL-1β treatment (Fig. 3C), as previously reported (Snelling et al, 2022). The TRAF6 ubiquitinylation observed in *H. pylori* infection is most likely auto-ubiquitinylation because overexpression of a dominant-negative catalytically inactive TRAF6 mutant (Fig. 3D), as well as pre-incubation of the cells with C25-140 (Fig. EV3C), significantly reduced TRAF6 ubiquitinylation. In addition, knockdown of cIAP1, another E3 ligase frequently involved in stimuli-induced NF-κB activation (Mahoney et al, 2008; Snelling et al, 2022), had virtually no effect on TRAF6 ubiquitinylation and NF-κB activation in response to *H. pylori* infection (Fig. 3E,F). The functionally redundant cIAP2 is upregulated only after 90 min of *H. pylori* infection (Fig. EV3D) and is therefore unlikely to play a role in the TRAF6 ubiquitinylation that we observed as early as 30 min post infection.

Based on these findings and in combination with our previous observations that implicate a non-catalytic mechanism by CYLD (Figs. 1I and EV3B), we hypothesized that the presence of CYLD stabilizes TRAF6 ubiquitinylation. A possible mechanism could be by counteracting the deubiquitinylase activity of A20, which was shown to catalyze the hydrolysis of polyubiquitin chain from TRAF6 (Boone et al, 2004). This mechanism is plausible because in *H. pylori* infection, we detected the interaction of A20 with a complex composed of TRAF6, CYLD, TIFA and TAK1 (Fig. 4A), likely via A20's association with TIFA (Lim et al, 2022) because recombinant A20 and CYLD did not interact directly (Fig. EV3E). We proceeded to use A20-knockout (A20^(KO)) cells (Lim et al, 2022) to assess whether A20 has an impact on early *H. pylori* infection-induced TRAF6 ubiquitinylation. Following *H. pylori* infection, A20^(KO) cells showed enhanced TRAF6 ubiquitinylation in a TRAF6 IP and NF-κB activation as indicated by the stronger

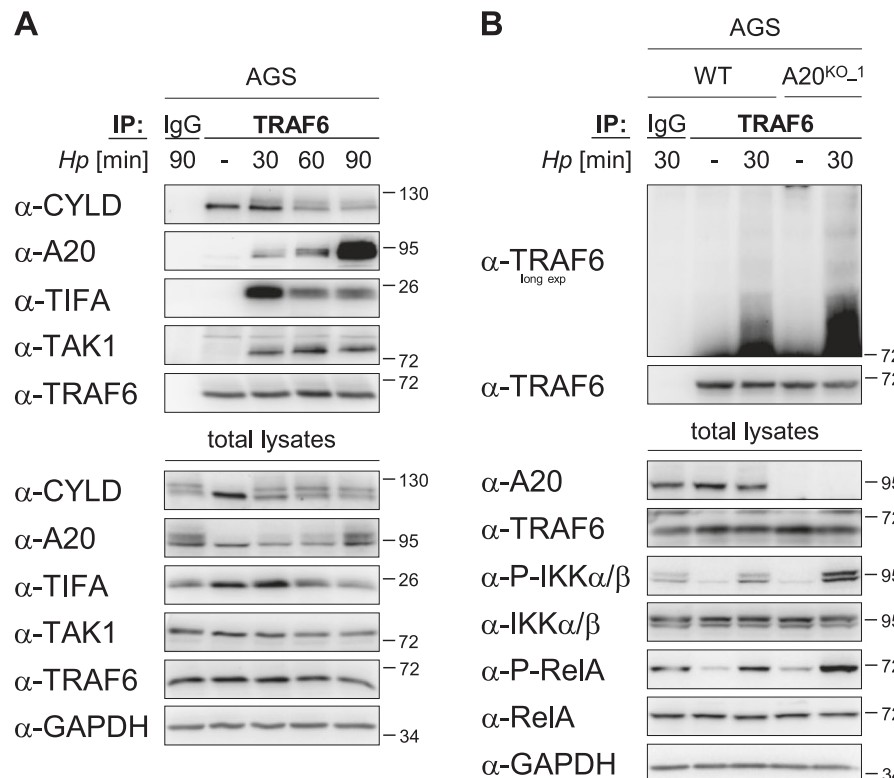

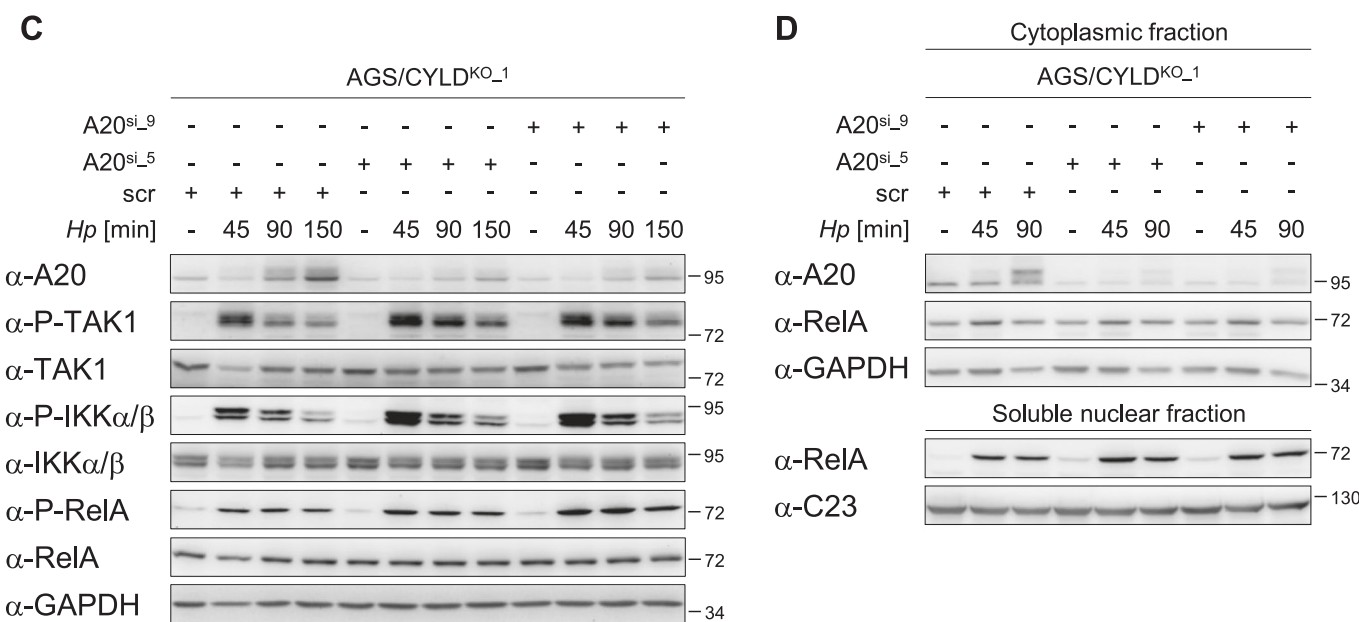

phosphorylation of IKKα/β and RelA in these cells compared to WT cells (Fig. 4B). To evaluate whether endogenous CYLD and A20 act counteractively, we decided to deplete A20 in CYLD^KO cells. Our reasoning was, if CYLD mitigates the activity of A20 during NF-κB activation, then depleting A20 in CYLD^KO cells should reverse the effect of CYLD deficiency on NF-κB activation. Indeed, upon *H. pylori* infection, NF-κB activation was stronger in CYLD^KO cells where A20 expression was knocked down using two independent siRNAs as

determined by the phosphorylation of TAK1, IKKα/β, and RelA (Fig. 4C) as well as the nuclear translocation of RelA (Fig. 4D).

## CYLD negatively influences the *H. pylori*-induced alternative NF-κB pathway through A20

As the classical NF-κB pathway was attenuated in *H. pylori*-infected CYLD^KO cells (Fig. 1A,B), we observed a corresponding decrease in the

**Figure 4. CYLD counterbalances TRAF6 deubiquitinylation by A20.**

(A) AGS cells were left uninfected (−) or infected with *H. pylori* for the times shown and harvested for total lysates. IP with an anti-TRAF6 antibody or an isotype-matched antibody (IgG) was performed. Eluates and total lysates were analyzed by immunoblotting using the indicated antibodies. (B) WT and A20^KO AGS cells were left uninfected or infected with *H. pylori* for 30 min and harvested for total lysates using lysis buffer containing 1% SDS (denaturing condition). IP with an anti-TRAF6 antibody or an isotype-matched antibody (IgG) was performed. Eluates and total lysates were analyzed by immunoblotting using the indicated antibodies. (C) AGS/CYLD^KO_1 cells were transfected with siRNAs targeting A20 (A20^si_5 or A20^si_9, 40 nM) or a non-targeting scrambled siRNA control (scr, 40 nM) 48 h prior to infection with *H. pylori* for the times shown. Total lysates were analyzed by immunoblotting using the indicated antibodies. (D) AGS/CYLD^KO_1 cells were transfected with siRNAs targeting A20 (A20^si_5 or A20^si_9, 40 nM) or a non-targeting scrambled siRNA control (scr, 40 nM) 48 h prior to infection with *H. pylori* for the times shown. Cell fractionation was performed to obtain the cytoplasmic and soluble nuclear fractions before analysis by immunoblotting using the indicated antibodies. GAPDH and C23 serve as the loading controls for the cytoplasmic and soluble nuclear fractions, respectively. For (A–C), GAPDH serves as the loading control for the total lysates. For all panels, IBs were processed in parallel and depict 1 representative of at least two independent experiments. Source data are available online for this figure.

classical NF-κB-dependent de novo synthesis of A20 (Fig. 5A). We showed previously that A20 perturbs the association of TIFA with the NIK regulatory complex, leading to inhibition of the alternative NF-κB pathway (Lim et al, 2022). Consistent with these earlier findings, we observed significant augmentation of the alternative NF-κB pathway in *H. pylori*-infected CYLD^KO cells, as shown by the accumulation of NIK as well as the nuclear translocation of p52 (Fig. 5A,B). The accumulation of NIK was also observed in ADP-heptose-treated CYLD^KO cells (Fig. EV4A). In LTα₁β₂-treated CYLD^KO cells, however, there was no accumulation of NIK since the abundance of A20 remained the same (Fig. EV4B). A20 and TIFA were found in TRAF2 immunoprecipitates 90 min after infection (Fig. 5C), where the presence of TIFA confirmed that A20 interacted with a TRAF2-containing entity that is involved in *H. pylori*-induced alternative NF-κB pathway (Maubach et al, 2021). Importantly, introducing recombinant human A20 protein into CYLD^KO cells led to a noticeable reduction in NIK accumulation upon *H. pylori* infection (Fig. 5D). Collectively, our data indicated that at later time points during *H. pylori* infection, an increased de novo synthesis of A20 leads to the termination of classical as well as alternative NF-κB.

In conclusion, we presented here data revealing a hitherto unexpected role of CYLD in supporting NF-κB activation in response to *H. pylori* infection. We show that CYLD interacts constitutively with TRAF6 and that upon *H. pylori* infection, this interaction stabilizes the ubiquitinylation of TRAF6. We also propose that A20 negatively regulates the ubiquitinylation of TRAF6 and its downstream signaling, and CYLD counters this effect, thus relieving the 'brakes' of A20 on classical NF-κB activation (Fig. 5E). Our findings contrast with previous reports (Kovalenko et al, 2003; Trompouki et al, 2003) addressing CYLD's DUB activity towards ubiquitinylated TRAF6. Both studies showed that transient overexpression of FLAG-tagged TRAF6 with HA-tagged ubiquitin resulted in TRAF6 polyubiquitinylation that was reduced when FLAG-tagged WT CYLD was co-transfected. It is therefore possible that these observations did not have physiological relevance due to CYLD overexpression.

Our data also indicate that the reduced NF-κB activation at later time points in *H. pylori*-infected CYLD^KO cells has a positive impact on the activation of alternative NF-κB due to the attenuated expression of A20. This is consistent with our findings from an earlier publication (Lim et al, 2022), where we showed that in *H. pylori* infection, A20 can stabilize the NIK regulatory complex, which is necessary for mediating the proteasomal degradation of NIK, providing a negative feedback loop that leads to shutdown of the alternative NF-κB pathway.

Studies that have reported on CYLD phosphorylation(s) have identified IKKβ (activated as part of the IKK complex due to receptor-mediated NF-κB activation) or IKKε as the kinases responsible (Elliott et al, 2021; Hutti et al, 2009; Reiley et al, 2005). We observed in our study that CYLD becomes phosphorylated in response to *H. pylori* infection. This phosphorylation seems to be strictly dependent on the ADP-heptose/ALPK1/TIFA axis because it was not observed in TIFA^KO cells. On the contrary, in cells transfected with siRNAs against TAK1, CYLD phosphorylation was still present. These findings suggest that NF-κB (ergo activated IKKβ) is not involved in *H. pylori*-induced CYLD phosphorylation.

In summary, our study demonstrates that CYLD enhances NF-κB signaling in response to *H. pylori* infection in a manner that differs from receptor-mediated NF-κB pathway. In view of this, it would also be interesting to study other reported functions of CYLD (Marín-Rubio et al, 2023), including cell death, cell cycle regulation and autophagy upon *H. pylori* infection.

# Methods

## Cell lines and culture conditions

AGS (ATCC® CRL-1739™), NCI-N87 (ATCC® CRL-5822™), A20^KO (Lim et al, 2022), TIFA^KO (Maubach et al, 2021), and CYLD^KO AGS

**Reagents and tools table**

| Reagent/resource | Reference or source | Identifier or catalog number |
|---|---|---|
| **Experimental models** | | |
| AGS | ATCC® | CRL-1739™ |
| NCI-N87 | ATCC® | CRL-5822™ |
| AGS/A20^KO | Lim et al, 2022 | |
| AGS/TIFA^KO | Maubach et al, 2021 | |

| Reagent/resource | Reference or source | Identifier or catalog number |
|---|---|---|
| AGS/CYLD^KO | This study | |
| NCI-N87/CYLD^KO | This study | |
| **Recombinant DNA** | | |
| HA-CYLD-WT | Addgene | 15506 |
| Myc-CYLD-C601A | Addgene | 60027 |
| pSELECT-puro-msc | Invivogen | psetp-mcs |
| pCMV3-TRAF6-FLAG (Fig. 2I) | Sino Biological Inc. | HG17419-CF |
| FLAG-TRAF6-wt | Addgene | 21624 |
| FLAG-TRAF6-C70A | This study | |
| pSELECT-puro-hCYLD-WT (1-953) | This study | |
| pSELECT-puro-hCYLD 1-220 | This study | |
| pSELECT-puro-hCYLD 1-309 | This study | |
| pSELECT-puro-hCYLD-301-580 | This study | |
| pSELECT-puro-hCYLD-301-953 | This study | |
| pSELECT-puro-hCYLD 581-953 | This study | |
| **Antibodies** | | |
| Rabbit α-A20<br>1:2000 dil in 5% BSA (Fig. EV3E) | Cell Signaling Technology | 5630 |
| Rabbit α-cIAP1<br>1:2000 dil in 5% BSA | Cell Signaling Technology | 7065 |
| Rabbit α-HA<br>(For IP) | Cell Signaling Technology | 3724 |
| Mouse α-HA<br>1:2000 dil in 5% milk (For IB) | Cell Signaling Technology | 2367 |
| Rabbit α-IKKα<br>1:2000 dil in 5% BSA | Cell Signaling Technology | 2682 |
| Rabbit α-IKKβ<br>1:2000 dil in 5% BSA | Cell Signaling Technology | 8943 |
| Rabbit α-IκBα<br>1:1000 dil in 5% BSA | Cell Signaling Technology | 4812 |
| Rabbit α-NF-κB2 p100/p52<br>1:2000 dil in 5% BSA | Cell Signaling Technology | 4882 |
| Rabbit α-NIK<br>1:3000 dil in 5% milk | Cell Signaling Technology | 4994 |
| Rabbit α-TAK1<br>1:2000 dil in 5% BSA | Cell Signaling Technology | 4505 |
| Rabbit α-TIFA<br>1:2000 dil in 5% BSA | Cell Signaling Technology | 61358 |
| Rabbit α-TRAF6<br>1:2000 dil in 5% BSA | Cell Signaling Technology | 8028 |
| Mouse α-phospho-IκBα<br>1:2000 dil in 5% milk | Cell Signaling Technology | 9246 |
| Rabbit α-phospho-IKKα/β<br>1:2000 dil in 5% BSA | Cell Signaling Technology | 2697 |
| Rabbit α-phospho-RelA<br>1:2000 dil in 5% BSA | Cell Signaling Technology | 3031 |
| Rabbit α-phospho-TAK1<br>1:2000 dil in 5% BSA | Cell Signaling Technology | 4508 |
| Mouse α-A20<br>1:1000 dil in 5% milk | Santa Cruz Biotechnology | sc-166692 |

| Reagent/resource | Reference or source | Identifier or catalog number |
|---|---|---|
| Rabbit α-C23 1:1000 dil in 5% milk | Santa Cruz Biotechnology | sc-13057 |
| Mouse α-CYLD 1:1000 dil in 5% milk | Santa Cruz Biotechnology | sc-74435 |
| Mouse α-RelA 1:1000 dil in 5% milk | Santa Cruz Biotechnology | sc-8008 |
| Mouse α-TRAF2 1:1000 dil in 5% milk | Santa Cruz Biotechnology | sc-136999 |
| Mouse α-ubiquitin 1:1000 dil in 5% milk | Santa Cruz Biotechnology | sc-8017 |
| Mouse α-FLAG 1:5000 dil in 5% milk | Sigma-Aldrich | F3165 |
| Mouse α-GAPDH 1:10,000 dil in 5% milk | Merck Millipore | MAB374 |
| Rabbit α-RNF31 (HOIP) 1:2000 dil in 5% milk | Novus Biologicals | NBP1-55059 |
| anti-rabbit-HRP 1:6000 dil in 5% milk | Jackson ImmunoResearch Laboratories | 711-036-152 |
| anti-mouse-HRP 1:6000 dil in 5% milk | Jackson ImmunoResearch Laboratories | 715-036-151 |
| anti-light chain-specific rabbit-HRP 1:6000 dil in 5% milk | Jackson ImmunoResearch Laboratories | 211-032-171 |
| **Oligonucleotides and other sequence-based reagents** | | |
| Alt-R® CRISPRCas9 guide RNA | IDT | Hs.Cas9.CYLD.1.AA |
| HA-tag hCYLD 1-220_s (5'-3') | AAATAAGGCGCGCCTC | This study |
| HA-tag hCYLD 1-220_as (5'-3') | CCCAGGACCTGCGTAATC | This study |
| HA-tag hCYLD 1-309_as (5'-3') | ACTCTCTGGGATGATATCATTG | This study |
| HA-tag hCYLD 581-end_s (5'-3') | AAAGAAGGCTTGGAGATAATGATTGGGAAG | This study |
| HA-tag hCYLD 581-end_as (5'-3') | GGGCCGGCCAGCGTA | This study |
| HA-tag hCYLD 1-300_s (5'-3') | GATATCATCCCAGAGAGTGTGACGC | This study |
| HA-tag hCYLD-301-580_as (5'-3') | GCCTTCTTTTTCCATTTTTGG | This study |
| s_CylD_BamHI (5'-3') | TTTGCAGGATCCACCATGTACC | This study |
| as_CylD_NheI (5'-3') | TCGTTGCTAGCAGAGGCGCGCCTTATTTGTAC | This study |
| hCYLD_E626A_for (5'-3') | TATAATATTCTACATCGTTCTTTGCTTTGGGTCTAAGTAACACAGTG | This study |
| hCYLD_E626A_rev (5'-3') | CACTGTGTTACTTAGACCCAAAGCAAAGAACGATGTAGAATATTATA | This study |
| msTRAF6C70A_FWD (5'-3') | CAAGTATGAGGCGCCCATTTGCTTGATGG | This study |
| msTRAF6C70A_REV (5'-3') | CTCTCCAGAGGTGGG | This study |
| **Chemicals, enzymes, and other reagents** | | |
| Recombinant human A20 protein | BPS Bioscience | 80408 |
| Recombinant human CYLD protein | OriGene | SKUTP319629 |
| Recombinant human TIFA protein | Novus Biologicals | NBP1-99103-50ug |
| Recombinant human TRAF6 protein | US Biological | 375664 |
| Recombinant human IL-1β | PeproTech | 200-01B-10UG |
| BamHI-HF | New England Biolabs GmbH | R3136S |
| NheI-HF | New England Biolabs GmbH | R3131S |
| T4 ligase | New England Biolabs GmbH | M0202 |
| Bovine serum albumin (BSA) | AppliChem GmbH | A1391 |
| Non-fat dry milk powder | CARL ROTH | T145.2 |

| Reagent/resource | Reference or source | Identifier or catalog number |
|---|---|---|
| **Software** | | |
| ChemoCam Imager | Intas | |
| StepOnePlus™ Real-Time PCR System | Applied Biosystems | |
| OriginPro 2020b | OriginLab Corporation | |
| **Other** | | |
| Lipofectamine® CRISPRMAX Transfection Reagent Kit | Thermo Fisher Scientific | CMAX00015 |
| NucleoSpin Gel and PCR Clean-up Kit | Macherey-Nagel | 740609.50 |
| NucleoSpin® RNA Plus Kit | Macherey-Nagel | 740984.250 |
| RevertAid™ First Strand cDNA Synthesis Kit | Thermo Fisher Scientific | K1621 |
| Q5® Site-Directed Mutagenesis Kit | New England Biolabs GmbH | E0554S |
| QuikChange XL Site-Directed Mutagenesis Kit | Agilent Technologies | 200516 |
| SensiMix™ SYBR® Hi-ROX Kit | Meridian Bioscience | QT605-05 |

cells were routinely cultured in RPMI-1640 medium (Gibco™, 21875–034) supplemented with 10% fetal calf serum (FCS) at 37 °C in a 5% $CO_2$ humidified incubator. For experiments, NCI-N87 cells were seeded at a density of $1 \times 10^6$ per 60 mm culture dish; other cells were seeded at a density of $6 \times 10^5$ per 60 mm culture dish or $2 \times 10^6$ per 100 mm culture dish. Cell culture medium was changed to RPMI-1640 medium supplemented with 0.2% FCS 16–20 h prior to infection with *H. pylori* or treatment with 10 ng/ml recombinant human IL-1β (PeproTech, 200-01B-10UG).

## Bacterial culture

The *H. pylori* WT P1 or P12 (used for Fig. EV1E only) strain was streaked from a −80 °C stock onto an agar plate containing 10% horse serum and 10 μg/ml vancomycin, and cultivated under microaerophilic conditions for 48–72 h. Bacteria were re-plated and cultivated for 24–48 h before use in experiments. The bacterial suspension used for infection of cells was prepared in phosphate-buffered saline (PBS) containing $Mg^{2+}$ and $Ca^{2+}$, and optical density at 550 nm was measured to determine the number of bacteria in the suspension. Infection of bacteria was performed at a multiplicity of infection (MOI) 100.

## Generation of CYLD-depleted AGS and NCI-N87 cell lines using CRISPR/Cas9

The knockout of CYLD in AGS or NCI-N87 cells was performed using the protocol according to Integrated DNA Technologies (IDT) as previously described (Maubach et al, 2021). The Alt-R® CRISPRCas9 guide RNA Hs.Cas9.CYLD.1.AA 5′-TCA CTG ACG GGG TGT ACC AA GGG-3′ (IDT) was used. Control AGS or NCI-N87 cells used have undergone the same CRISPR/Cas9 procedure but were not successful for the depletion of CYLD.

## Generation of mouse gastric spheroids

The stomachs of male C57BL/6 mice were provided by Dr. Verena Keitel-Anselmino (Clinic of Gastroenterology, Hepatology, and Infectiology, Medical Faculty, Otto-von-Guericke University Magdeburg), and the stomachs of male C57BL/6 mice homozygous

for *Cyld*$^{-/-}$ were provided by Dr. Ari Waisman (Institute for Molecular Medicine, University Medical Center of the Johannes Gutenberg-University Mainz). The mice were handled by trained and authorized personnel. The mice had access to water and food ad libitum, and were maintained under 12-hour light/dark cycles. The organ removal experiments were approved by the local authorities and the animal welfare officer of the local animal facility (permit numbers: KGHI-KEI-TWZ-1-23 and TVA G17-1-093).

Mouse spheroids were generated from isolated antral gastric gland units from stomachs of C57BL/6 WT or *Cyld*$^{-/-}$ mice as previously described (Bartfeld and Clevers, 2015). The isolated gastric glands were mixed with 50 μl Matrigel (Corning® Matrigel® Basement Membrane Matrix, LDEV-free, 354234) and plated in pre-warmed 24-well plates. After polymerization of the Matrigel, each well was overlaid with 500 μl warm culture medium (Adv. D-MEM/F12, Thermo Fisher Scientific, 12634-010) supplemented with 10 mM HEPES (Thermo Fisher Scientific, 15630056), 1% Glutamax (Thermo Fisher Scientific, 35050038), 1% (v/v) penicillin/streptomycin (Thermo Fisher Scientific, 15140-122), 2% B27 supplement (Thermo Fisher Scientific, 17504-044), 1% $N_2$ supplement (Thermo Fisher Scientific, 17502-048), 20 ng/ml human epidermal growth factor (EGF) (Thermo Fisher Scientific, PHG0311), 25% conditioned R-spondin medium, 25 ng/ml Wnt surrogate-Fc fusion protein (ImmunoPrecise Antibodies Ltd, N001-0.1 mg), 150 ng/ml human noggin (Peprotech, 120-10C), 150 ng/ml human fibroblast growth factor (FGF)-10 (Peprotech, 100-26), 1.25 mM N-acetyl-L-cysteine (Sigma-Aldrich, A9165), 10 mM nicotinamide (Sigma-Aldrich, N0636-100G), 10 nM human [Leu$^{15}$]-Gastrin I (Sigma-Aldrich, G9145), 7.5 μM ROCK inhibitor (Y-27632) (Sigma-Aldrich, Y0503) and 1 μM TGF-β RI Kinase Inhibitor IV (Alk-I) (Calbiochem, 616454), and incubated at 37 °C, 5% $CO_2$ in a humidified incubator. The culture medium was changed every 2–4 days and spheroids were passaged when necessary (depending on density). For experiments, WT and *Cyld*$^{-/-}$ spheroids were seeded at 20,000 cells per 50 μl Matrigel dome in 500 μl culture medium for 3–4 days and then treated with 500 nM ADP-heptose or 10 ng/ml recombinant human IL-1β for different times. Spheroids were harvested in ice-cold PBS, recovered from Matrigel by centrifugation, and lysed in cell lysis buffer.

**A**

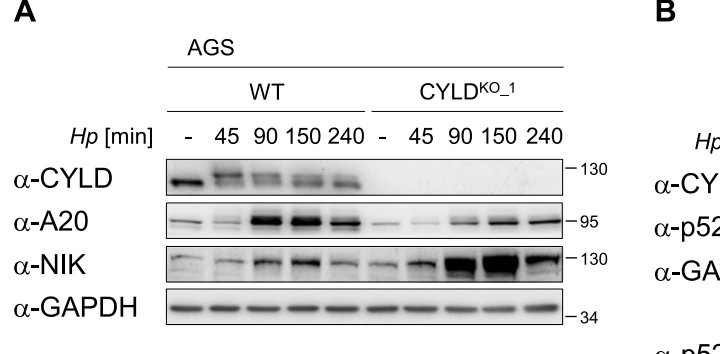

**B**

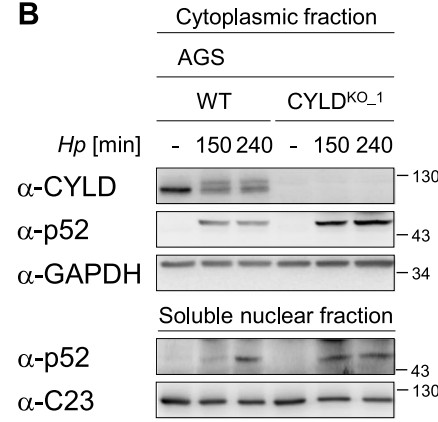

**C**

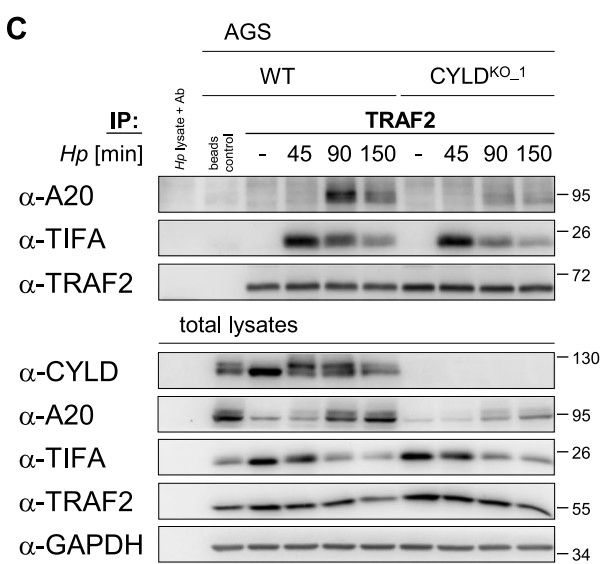

**D**

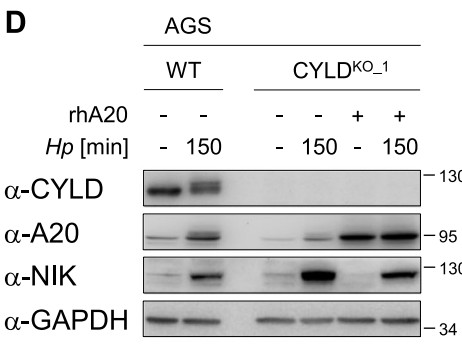

**E**

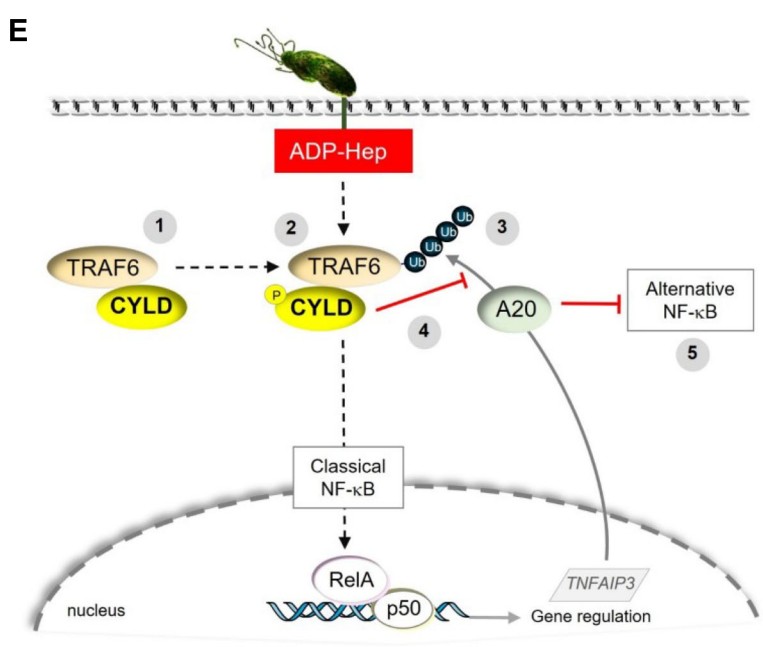

**Figure 5. CYLD negatively impacts *H. pylori*-induced alternative NF-κB activation through A20.**

(**A**) WT and CYLD$^{KO-1}$ AGS cells were left uninfected (-) or infected with *H. pylori* for the times shown. Total lysates were analyzed by immunoblotting using the indicated antibodies. (**B**) WT and CYLD$^{KO-1}$ AGS cells were left uninfected or infected with *H. pylori* for the times shown. Cell fractionation was performed to obtain the cytoplasmic and soluble nuclear fractions before analysis by immunoblotting using the indicated antibodies. GAPDH and C23 serve as the loading controls for the cytoplasmic and soluble nuclear fractions, respectively. (**C**) WT and CYLD$^{KO-1}$ AGS cells were left uninfected or infected with *H. pylori* for the times shown and harvested for total lysates. IP with an anti-TRAF2 antibody was performed. Lane 1, TRAF2 IP using lysate from *H. pylori* (*Hp* lysate + Ab) and lane 2, total lysates from *Hp* 150 min treatment plus protein A/G magnetic beads (beads control) indicate the specificity of the IP and serve as controls. Eluates and total lysates were analyzed by immunoblotting using the indicated antibodies. (**D**) WT and CYLD$^{KO-1}$ AGS cells were left uninfected or infected with *H. pylori* for 150 min. CYLD$^{KO-1}$ AGS cells were transfected with recombinant human (rh) A20 proteins 24 h prior to infection by *H. pylori* for 150 min. Total lysates were analyzed by immunoblotting using the indicated antibodies. (**E**) A proposed model for CYLD's role in the regulation of NF-κB signaling pathway in *H. pylori* infection. CYLD interacts constitutively with TRAF6, and upon *H. pylori* infection, this interaction stabilizes the ubiquitinylation of TRAF6. A20 negatively regulates the ubiquitinylation of TRAF6 and the ensuing classical NF-κB signaling. CYLD counters this effect, thus relieving the 'brakes' of A20 on classical NF-κB activation. In addition, the subsequent classical NF-κB-dependent de novo synthesis of A20 provides a negative feedback loop leading to shutdown not only of the classical but also of the alternative NF-κB pathway. For (**A, C, D**), GAPDH serves as the loading control for the total lysates. For (**A–D**), IBs were processed in parallel and depict 1 representative of at least two independent experiments. Source data are available online for this figure.

## Plasmids construction

The plasmid HA-CYLD-WT was a gift from Stephen Elledge (Addgene plasmid #15506; http://n2t.net/addgene:15506; RRID:Addgene_15506 (Stegmeier et al, 2007)). For stable transfection, the vector pSELECT-puro-msc (Invivogen, psetp-mcs) was chosen for puromycin selection. WT CYLD was amplified from HA-CYLD-WT using sense and antisense primers containing BamHI and NheI restriction sites, respectively (listed in the "Reagents and Tools Table"). The PCR product and the plasmid pSELECT-puro-msc were digested with BamHI/NheI, followed by gel extraction. Ligation was performed using T4 ligase at a ratio of 3:1 (insert:plasmid) for 1 h at room temperature. The ligation product (pSELECT-hCYLD-WT) was used to transform chemically competent *E. coli* NEB® 5-alpha. The transformed bacteria were plated on LB medium containing 100 μg/ml puromycin and cultured for 16 h. Selected positive colonies were confirmed by whole plasmid sequencing using Oxford Nanopore technology (Eurofins).

Mutagenesis of the TRAF6-binding motif E626A was performed using the Agilent QuikChange XL Site-Directed Mutagenesis kit on the plasmid pSELECT-hCYLD-WT using sense and antisense primers listed in the "Reagents and Tools Table". Puromycin-selected positive colonies were confirmed by whole plasmid sequencing using Oxford Nanopore technology (Eurofins).

The plasmid FLAG-TRAF6-WT was a gift from John Kyriakis, (Addgene plasmid #21624; http://n2t.net/addgene:21624; RRID:Addgene_21624, (Zhong and Kyriakis, 2004)). The E3 ubiquitin ligase activity-deficient plasmid FLAG-TRAF6-C70A was generated from FLAG-TRAF6-WT. We used the Q5® Site-Directed Mutagenesis kit to generate five truncated variants of human CYLD isoform 2 (missing aa 305–307) from the pSELECT-puro-hCYLD-WT plasmid, utilizing primers designed by NEBaseChanger® (listed in the "Reagents and Tools Table"). The five truncated CYLD variants (numbering according to isoform 2) are: aa 1–220, 1–309, 301–580, 301–953, 581–953. All generated plasmids were digested with BamHI/NheI for verification (Fig. EV2D) and subjected to sequencing using Oxford Nanopore technology (Eurofins).

## Transfection of siRNAs, plasmids, and recombinant proteins

Transfection of siRNAs was performed using siLentFect™ (Bio-Rad, 1703362) according to the manufacturer's protocol. One day

before siRNA transfections, cells were seeded at a density of $5 \times 10^5$ per 60 mm culture dish or $1.5 \times 10^6$ per 100 mm culture dish. Cell culture medium was changed to Opti-MEM (Gibco™, 31985070) prior to transfection. The siRNAs were used at a final concentration of 40 nM for A20 and CYLD, 25 nM for cIAP1 and 20 nM for TAK1. As a control for the unspecific effects of the transfection procedure, equimolar concentration of scrambled (non-targeting) siRNA was used for the respective experiments. The medium was changed to RPMI-1640 medium containing 10% FCS 6 h after transfection. After incubation for 16–20 h, the cell culture medium was changed to RPMI-1640 medium supplemented with 0.2% FCS. After incubation for 16–20 h, the cells were infected with *H. pylori*. The following siRNAs were used: scrambled siRNA (Qiagen, SI03650318), A20$^{si-5}$ (Dharmacon, J-009919–05-0005), A20$^{si-9}$ (Qiagen, SI05018601), cIAP1$^{si}$ (Qiagen, SI02654442), CYLD$^{si\_E1}$ (Eurofins, 5′-GAUUGUUACUUCUAUCAAA-3′), CYLD$^{si\_UTR}$ (Eurofins, 5′-GCAGAGTCCTAACGTTGCA-3′ (Stegmeier et al, 2007)), HOIP (Dharmacon, M-021419-01-0010) and TAK1$^{si\_E1}$ (Eurofins, 5′-AUUUCAGACAUGUCAGCAC-3′).

The plasmids HA-CYLD-WT and HA-CYLD-C601A were used for transient transfection. HA-CYLD-C601A was a gift from Stephen Elledge (Addgene plasmid #60027; http://n2t.net/addgene:60027; RRID:Addgene_60027, (Stegmeier et al, 2007)). After whole plasmid sequencing, we realized that the purchased HA-CYLD-C601A did not contain the expected N-terminal 3x HA-Tag but a longer N-terminal sequence containing 6x Myc-Tag, amounting to about 10 kDa. We have informed Addgene, following which the name of the plasmid was changed to Myc-CYLD-C601A.

Cells were seeded at a density of $1.2 \times 10^6$ per 60 mm culture dish 24 h prior to transfection. The amount of plasmid used per 60-mm culture dish was 6 μg. Plasmid transfection was performed using METAFECTENE® PRO transfection reagent (Biontex Laboratories, T040-5.0) according to the suggested protocol from the manufacturer for the 60 mm culture dish format. The ratio of plasmid (μg) to transfection reagent (μl) used was 1:5. The cell culture medium was changed to RPMI-1640 medium containing 10% FCS before transfection. The plasmids were transfected into the cells for 24 h followed by *H. pylori* infection.

For the generation of stable clones of CYLD$^{KO-1}$ AGS cells with pSelect empty vector or WT CYLD, the same transfection protocol as above was used with subsequent Zeocin selection using the limited dilution method.

Transfection of recombinant human A20 protein was performed using reagents from the Lipofectamine® CRISPRMAX transfection reagent kit. Twenty-four hours prior to protein transfection, cells were seeded at a density of $7.5 \times 10^5$ per 60 mm culture dish. The cell culture medium was changed to RPMI-1640 medium supplemented with 0.2% FCS prior to transfection. One microgram recombinant protein was combined with 5 µl Cas9 PLUS™ reagent in 500 µl Opti-MEM and incubated at room temperature for 5 min. Six microliters of CRISPRMAX transfection reagent diluted in 500 µl Opti-MEM was combined with the diluted Cas9 PLUS™ reagent/recombinant protein solution and incubated at room temperature for 20 min before adding drop-wise to the cells in the dishes. After incubation for 16–20 h, the cells were infected with *H. pylori*. The method described applies to the transfection of protein for cells in one 60 mm culture dish.

## SDS-PAGE and immunoblotting

For preparation of total cell lysates, cells were washed twice in ice-cold PBS and scraped in 100 µl (for 60 mm cell culture dish) or 750 µl (for 100 mm cell culture dish) lysis buffer (50 mM Tris/HCl pH 7.5, 150 mM NaCl, 5 mM EDTA, 10 mM $K_2HPO_4$, 10% glycerol, 1% Triton X-100 and 0.5% NP-40) containing phosphatase inhibitors (1 mM sodium vanadate, 1 mM sodium molybdate, 20 mM sodium fluoride, 10 mM sodium pyrophosphate, 1 mM AEBSF and 20 mM 2-phosphoglycerate) and protease inhibitor mix (Roche Diagnostics, 34044100). Lysates were transferred into microtubes, vortexed briefly, and incubated on ice for 15 min. For the preparation of cell lysates for IP, the cell lysates were sheared through a 26 G x 1/2" needle several times prior to incubation on ice. Cleared cell lysates were obtained after centrifugation at 13,000 rpm for 15 min at 4 °C.

For the preparation of cell lysates under denaturing conditions, cells were lysed in lysis buffer containing phosphatase and protease inhibitors (as described in the previous paragraph), as well as 1% SDS, and sheared by passing several times through a 26 G x 1/2" needle attached to a 1 ml syringe. The cell suspensions were heated at 95 °C for 10 min followed by centrifugation at 13,000 rpm for 15 min. The cleared cell lysates were diluted 1:10 in lysis buffer (without phosphatase inhibitors, protease inhibitors, and SDS) before use for IP.

The subcellular fractionation of cells into cytosolic and soluble nuclear fractions was performed as described in previous work (Studencka-Turski et al, 2018).

Protein concentration was determined using the BCA protein assay kit (Thermo Fisher Scientific, 23225). About 10–25 µg of cell lysates were supplemented with 1× sample buffer (50 mM Tris-HCl pH 6.8, 2% SDS, 5% glycerol, 2% 2-mercaptoethanol and 0.02% bromophenol blue), heated at 95 °C for 10 min, separated in 6, 9, or 10% polyacrylamide gels and transferred onto PVDF membranes (Merck Millipore). The membranes were blocked for 1 h at room temperature using 5% skim milk in TBS containing 0.1% Tween (TBS-T). The membranes were incubated overnight with primary antibodies at the appropriate dilutions in either 5% BSA or 5% skim milk in TBS-T at 4 °C on a rocking platform. The membranes were washed three times in TBS-T and incubated with the appropriate HRP-conjugated secondary antibody for 1 h at room temperature at a dilution of 1:6000 in 5% skim milk in TBS-T, followed by three washes in TBS-T. The membranes were developed using the chemiluminescent HRP substrate (Merck Millipore, WBKLS0500 or Cytiva, RPN2209, for detection of GAPDH only). The band pattern was visualized using the ChemoCam Imager (Intas). All antibodies used in this work are reported in the "Reagents and Tools Table". The secondary anti-light chain-specific rabbit-HRP antibody was used for analysis of immunoblots after IPs.

## Immunoprecipitation

Two milligrams of total cell lysates were incubated with 0.5 µg α-TRAF6, 10 µg α-TIFA or isotype-matched IgG overnight. Pre-washed Protein A/G magnetic beads (Thermo Fisher Scientific, 88803) were added, followed by incubation for 1 h. Both incubation steps were carried out at 4 °C on a rotator at 7 rpm. The beads with immunoprecipitates were washed four times in lysis buffer (without protease and phosphatase inhibitors) and once in PBS. Elution of immunoprecipitated proteins from the beads was achieved by incubation with 2× sample buffer for 20 min at room temperature. Eluate was transferred to a clean microtube and heated at 95 °C for 10 min prior to SDS-PAGE and immunoblotting.

For the experiments in Fig. 2D–F, 50 ng recombinant human c-Myc-FLAG-tagged CYLD (OriGene, SKUTP319629), His-tagged TIFA (Novus Biologicals, NBP1-99103-50ug) or His-SUMO-tagged TRAF6 (US Biological, 375664) were combined in 1 ml PBS containing $Mg^{2+}$, $Ca^{2+}$, and 150 mM NaCl, and incubated for 1 h. IP was performed by incubation with 5 µg α-CYLD, 10 µg α-TIFA or 0.5 µg α-TRAF6 antibody for 1 h. Both incubation steps were carried out at 4 °C on a rotator at 7 rpm. The addition of pre-washed Protein A/G magnetic beads and subsequent steps were the same as described in the previous paragraph.

## Quantitative PCR

Total RNA was isolated using the NucleoSpin® RNA Plus kit. Five micrograms of total RNA were reverse transcribed into cDNA using the RevertAid™ First Strand cDNA Synthesis kit. Quantitative PCR was performed using the primer sets targeted against *TNF*, *IL-8*, and *RPL13* from RealTimePrimers (VHPS-9415, VHPS-4563, and VHPS-7964, respectively). Quantitative PCR was performed using the StepOnePlus™ Real-Time PCR System (Applied Biosystems, Thermo Fisher Scientific). Data were normalized to the *RPL13* housekeeping gene and expressed as fold change in mRNA expression relative to uninfected cells (Comparative CT Method (ΔΔCT)). Triplicate determinations of each experiment were performed.

## Statistical analysis

Quantitative PCR data were analyzed using OriginPro 2020b and are presented as means ± SD. The significance of the quantitative data was tested using the parametric unpaired two-sample Student's *t* test. $P < 0.05$ was regarded as statistically significant.

# Data availability

No data were deposited in a public database.

The source data of this paper are collected in the following database record: biostudies:S-SCDT-10_1038-S44319-025-00480-y.

## Peer review information

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

## Acknowledgements

We thank Dr. M Reich (Otto von Guericke University Magdeburg) and Y. Kattimani (Johannes Gutenberg University Mainz) for their support in our work involving mice. This work was supported in part by the European Union Program European Regional Development Fund of the Ministry of Economy, Science and Digitalisation in Saxony-Anhalt within the Center of Dynamic Systems (ZS/2023/12/182075) and by a grant of the German Research Foundation (Na 292/17-1) to MN.

## Author contributions

**Michelle C C Lim**: Conceptualization; Investigation; Writing—original draft; Writing—review and editing. **Gunter Maubach**: Investigation; Writing—review and editing. **Michael Naumann**: Conceptualization; Supervision; Funding acquisition; Visualization; Writing—original draft; Project administration; Writing—review and editing.

Source data underlying figure panels in this paper may have individual authorship assigned. Where available, figure panel/source data authorship is listed in the following database record: biostudies:S-SCDT-10_1038-S44319-025-00480-y.

## Funding

## Disclosure and competing interests statement

The authors declare no competing interests.

# Expanded View Figures

**Figure EV1.  CYLD supports *H. pylori*-induced classical NF-κB activation.**

(A) WT and CYLDKO_2 AGS cells were left uninfected (-) or infected with *H. pylori* for the times shown. Total lysates were analyzed by immunoblotting using the indicated antibodies. (B) WT and CYLDKO_2 AGS cells were left uninfected or infected with *H. pylori* for the times shown. Cell fractionation was performed to obtain the cytoplasmic and soluble nuclear fractions before analysis by immunoblotting using the indicated antibodies. GAPDH and C23 serve as the loading controls for the cytoplasmic and soluble nuclear fractions, respectively. (C) AGS cells were transfected with siRNAs targeting CYLD (CYLDsi_E1, 40 nM) or a non-targeting scrambled siRNA control (scr, 40 nM) 48 h prior to infection with *H. pylori* for the times shown. Total lysates were analyzed by immunoblotting using the indicated antibodies. (D) AGS cells were transfected with siRNAs targeting CYLD (CYLDsi_UTR, 40 nM) or a non-targeting scrambled siRNA control (scr, 40 nM) 48 h prior to infection with *H. pylori* for the times shown. Total lysates were analyzed by immunoblotting using the indicated antibodies. (E) WT and CYLDKO_1 AGS cells were left uninfected or infected with *H. pylori* WT P12 strain for the times shown. Total lysates were analyzed by immunoblotting using the indicated antibodies. (F) WT and CYLDKO_1 NCI-N87 cells were left uninfected or infected with *H. pylori* for the times shown. Total lysates were analyzed by immunoblotting using the indicated antibodies. (G) AGS cells were left uninfected or infected with *H. pylori* for 45 min. The *H. pylori*-infected sample was incubated with or without λ phosphatase in the appropriate reaction buffer. Total lysates were analyzed by immunoblotting using the indicated antibodies. (H) WT and CYLDKO_2 AGS cells were left untreated or treated with 10 ng/ml IL-1β for the indicated times. Total lysates were analyzed by immunoblotting using the indicated antibodies. (I) Antral gastric spheroids of two different pairs of mice were left untreated or treated with 500 nM ADP-heptose for the times shown. Total lysates were analyzed by immunoblotting using the indicated antibodies. (J) AGS cells were transfected with siRNAs targeting HOIP (HOIPsi, 30 nM) or a non-targeting scrambled siRNA control (scr, 30 nM) 48 h prior to infection with *H. pylori* for the times shown. Total lysates were analyzed by immunoblotting using the indicated antibodies. For (A, C–J), GAPDH serves as the loading control for the total lysates. For all panels, IBs were processed in parallel and depict 1 representative of at least two independent experiments. Asterisk denotes an unspecific band. Source data are available online for this figure.

▶

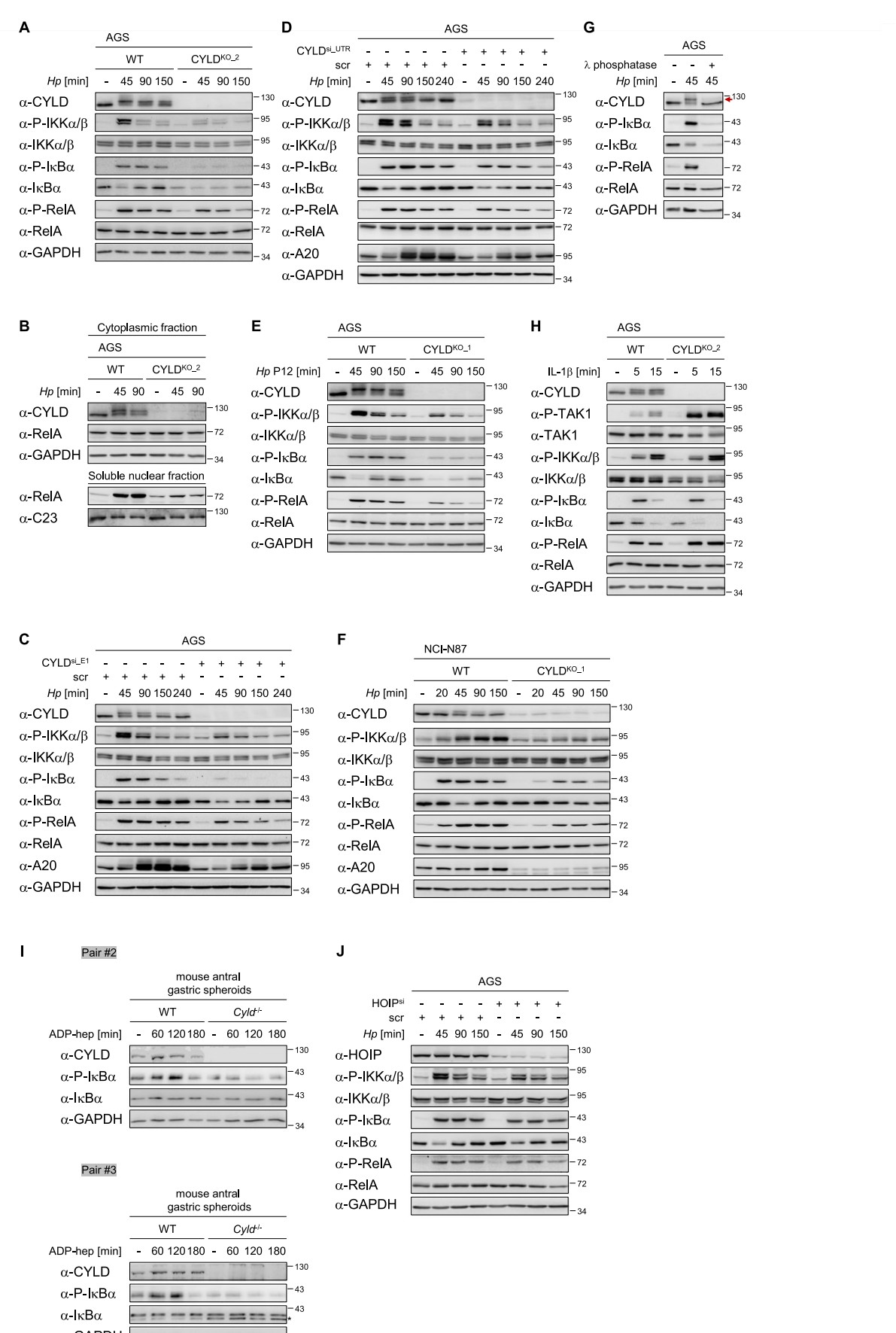

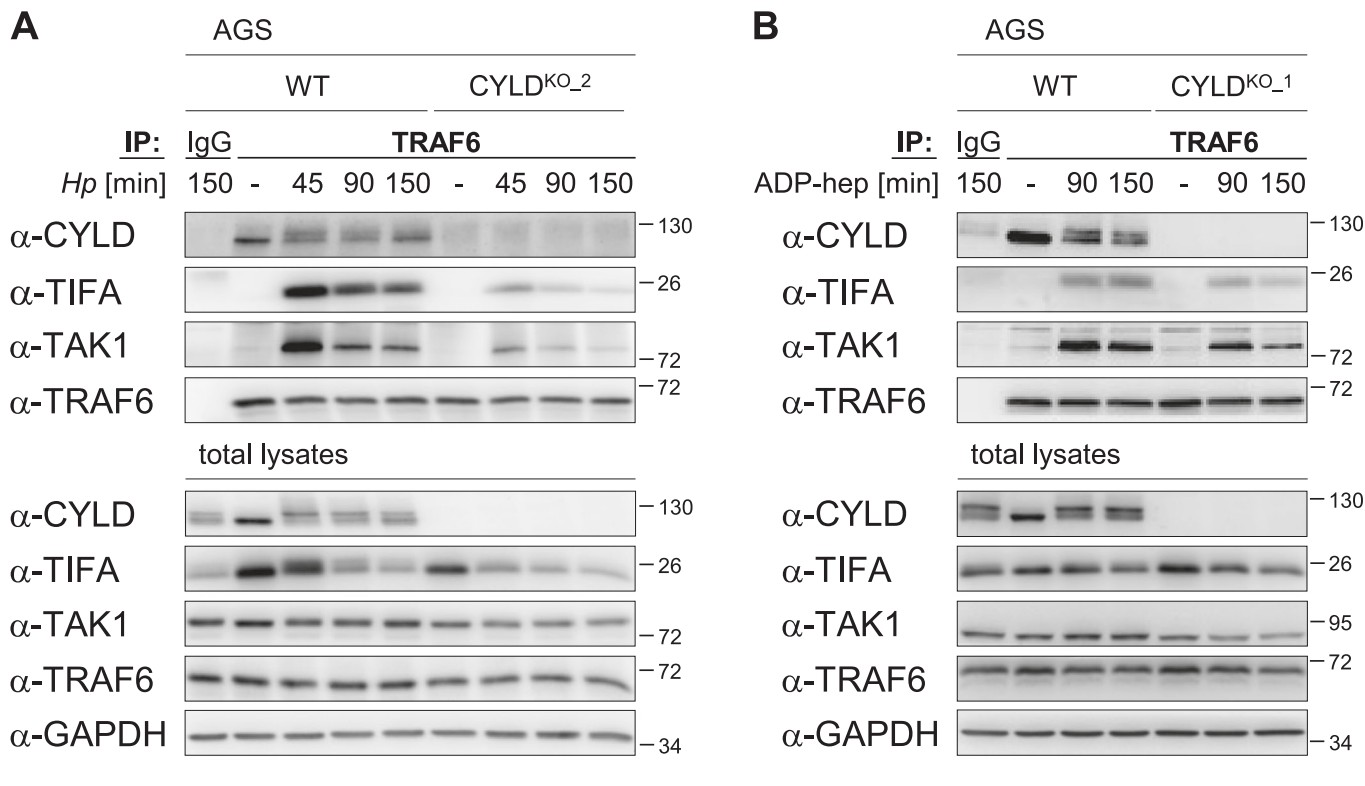

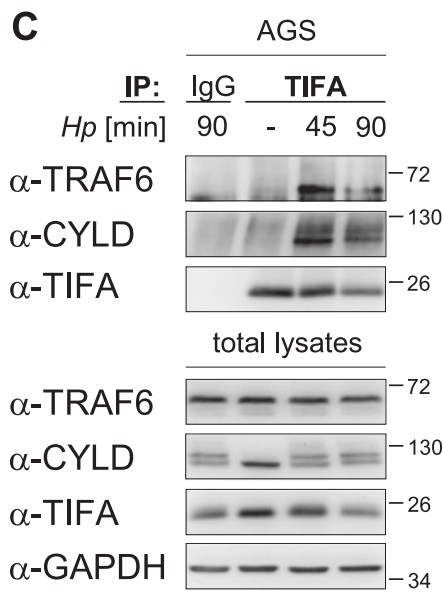

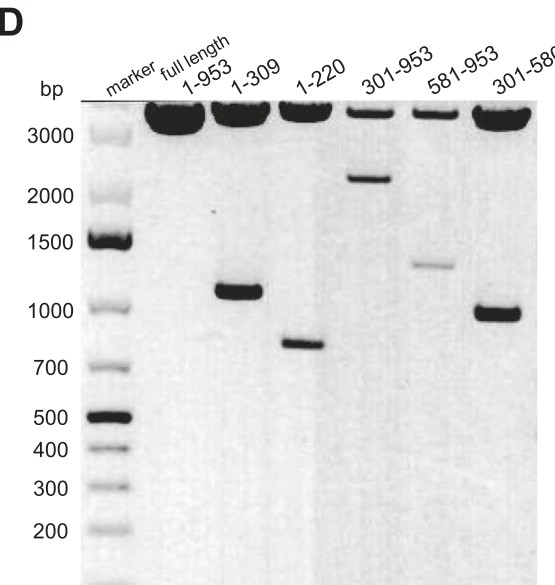

**Figure EV2. CYLD associates with the TIFA/TRAF6/TAK1 complex upon *H. pylori* infection.**

(A) WT and CYLD^KO_2 AGS cells were left uninfected (−) or infected with *H. pylori* for the times shown and harvested for total lysates. IP with an anti-TRAF6 antibody or isotype-matched antibody (IgG) was performed. Eluates and total lysates were analyzed by immunoblotting using the indicated antibodies. (B) WT and CYLD^KO_1 AGS cells were left untreated or treated with 200 nM ADP-heptose for the times shown and harvested for total lysates. IP with an anti-TRAF6 antibody or isotype-matched antibody (IgG) was performed. Eluates and total lysates were analyzed by immunoblotting using the indicated antibodies. (C) AGS cells were left uninfected or infected with *H. pylori* for the times shown and harvested for total lysates. IP with an anti-TIFA antibody or isotype-matched antibody (IgG) was performed. Eluates and total lysates were analyzed by immunoblotting using the indicated antibodies. (D) Full-length and the five truncated variants of human CYLD isoform 2 were digested with BamHI/NheI und analyzed on a 1% agarose gel containing GelRed®. The expected fragment sizes for the restriction digests of the full-length and CYLD variants were (in bp): full-length: 3366, 2975; 1–309: 3366, 1037; 1–220: 3366, 779; 301–953: 3366, 2075; 581–953: 3366, 1244; 301–580: 3366, 959. For (A–C), IBs were processed in parallel and depict 1 representative of at least two independent experiments. Source data are available online for this figure.

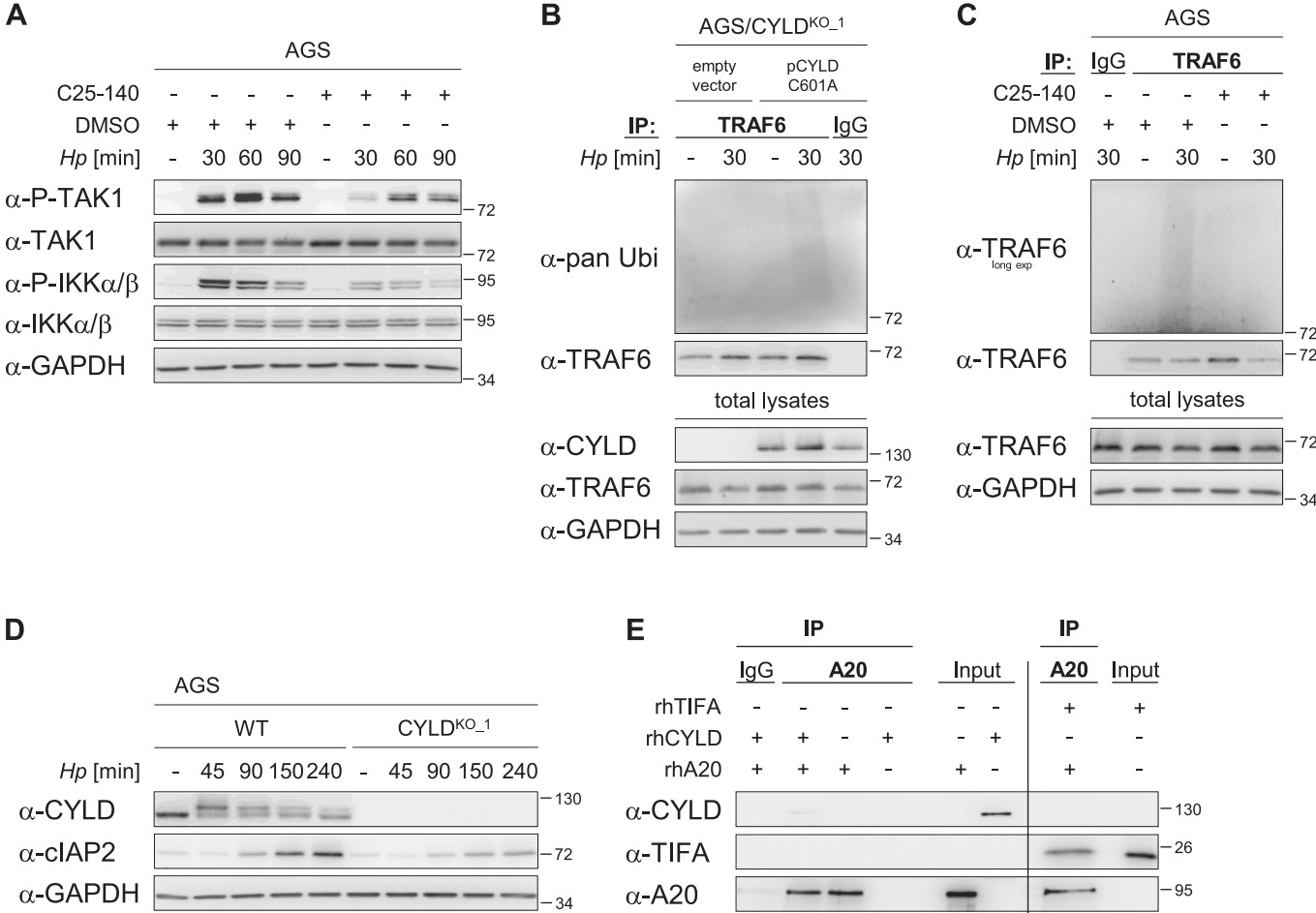

Figure EV3.   CYLD stabilizes TRAF6 ubiquitinylation.

(A) AGS cells were incubated with DMSO (vehicle) or a selective TRAF6-Ubc13 inhibitor (C25-140, 20 μM) for 6 h followed by *H. pylori* infection for the times shown. Total lysates were analyzed by immunoblotting using the indicated antibodies. (B) CYLD[KO_1] AGS cells were transfected with empty vector or plasmid expressing the catalytically inactive CYLD mutant (pCYLD C601A) 48 h prior to infection with *H. pylori* for 30 min. Total lysates were harvested using lysis buffer containing 1% SDS (denaturing condition). IP was performed using an anti-TRAF6 antibody or isotype-matched antibody (IgG). Eluates and total lysates were analyzed by immunoblotting using the indicated antibodies. (C) AGS cells were incubated with DMSO (vehicle) or a selective TRAF6-Ubc13 inhibitor (C25-140, 20 μM) for 6 h followed by *H. pylori* infection for 30 min. Total lysates were harvested using lysis buffer containing 1% SDS (denaturing condition). IP was performed using an anti-TRAF6 antibody or isotype-matched antibody (IgG). Eluates and total lysates were analyzed by immunoblotting using the indicated antibodies. (D) AGS cells were left uninfected (-) or infected with *H. pylori* for the times shown. Total lysates were analyzed by immunoblotting using the indicated antibodies. (E) Following incubation of recombinant human (rh) CYLD and rhA20 proteins or rhTIFA and rhA20 in vitro, IP was performed using an antibody against A20. Eluates and input (10 ng rhCYLD, rhA20 or rhTIFA proteins) were analyzed by immunoblotting using the indicated antibodies. For (A–D), GAPDH serves as the loading control for the total lysates. IBs were processed in parallel and depict 1 representative of at least two independent experiments. Source data are available online for this figure.

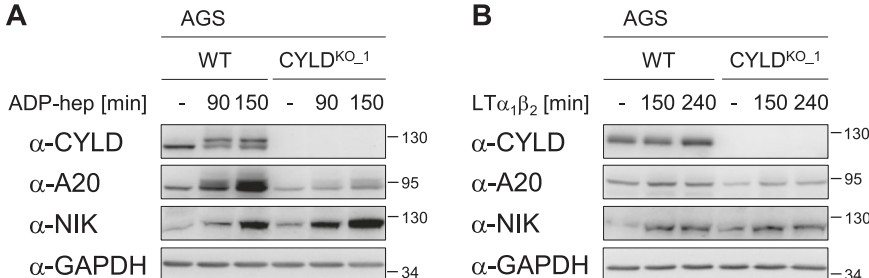

**Figure EV4.  CYLD negatively impacts *H. pylori*-induced alternative NF-κB activation through A20.**

(A) WT and CYLD[KO_1] AGS cells were left untreated (-) or treated with 200 nM ADP-heptose for the times shown. Total lysates were analyzed by immunoblotting using the indicated antibodies. (B) WT and CYLD[KO_1] AGS cells were left untreated or treated with 30 ng/ml LTα1β2 for the times shown. Total lysates were analyzed by immunoblotting using the indicated antibodies. For both panels, GAPDH serves as the loading control for the total lysates. IBs were processed in parallel and depict 1 representative of at least two independent experiments. Source data are available online for this figure.

