## [Peer Review File · EMBO Reports]

CYLD-TRAF6 interaction promotes ADP-heptose-induced NF- κ B signaling in *H. pylori* infection

Michelle CC Lim, Gunter Maubach, and Michael Naumann

Corresponding author(s): Michael Naumann (naumann@med.ovgu.de)

Review Timeline:

Submission Date:	22nd Nov 24
Editorial Decision:	2nd Jan 25
Revision Received:	1st Apr 25
Editorial Decision:	29th Apr 25
Revision Received:	8th May 25
Accepted:	12th May 25

Editor: Achim Breiling

Transaction Report:

Dear Prof. Naumann,

Thank you for the submission of your manuscript to EMBO reports. I have now received the reports from the three referees that were asked to evaluate your study, which can be found at the end of this email.

As you will see, the referees think that these findings are of interest. However, they have several comments, concerns, and suggestions, indicating that a major revision of the manuscript is necessary to allow publication of the study in EMBO reports. As the reports are below, and all the referee concerns need to be addressed, I will not detail them here.

Given the constructive referee comments, I would like to invite you to revise your manuscript with the understanding that the concerns of the referees must be addressed in the revised manuscript and in a detailed point-by-point response. Acceptance of your manuscript will depend on a positive outcome of a second round of review. It is EMBO reports policy to allow a single round of revision only and acceptance of the manuscript will therefore depend on the completeness of your responses included in the next, final version of the manuscript.

- 1) a .docx formatted version of the final manuscript text (including legends for main figures, EV figures and tables), but without the figures included. Figure legends should be compiled at the end of the manuscript text.
- 2) individual production quality figure files as .eps, .tif, .jpg (one file per figure), of main figures and EV figures. Please upload these as separate, individual files upon re-submission.

- 4) a complete author checklist, which you can download from our author guidelines (<https://www.embopress.org/page/journal/14693178/authorguide>). Please insert page numbers in the checklist to indicate where the requested information can be found in the manuscript. The completed author checklist will also be part of the RPF.

- 5) that primary datasets produced in this study (e.g. RNA-seq, ChIP-seq, structural and array data) are deposited in an

appropriate public database. If no primary datasets have been deposited, please also state this in a dedicated section (e.g. 'No primary datasets have been generated and deposited'), see below.

The accession numbers and database should be listed in a formal "Data Availability" section that follows the model below. This is now mandatory (like the COI statement). Please note that the Data Availability Section is restricted to new primary data that are part of this study. This section is mandatory. As indicated above, if no primary datasets have been deposited, please state this in this section

Data availability

8) Regarding data quantification and statistics, please make sure that the number "n" for how many independent experiments were performed, their nature (biological versus technical replicates), the bars and error bars (e.g. SEM, SD) and the test used to calculate p-values is indicated in the respective figure legends (also for EV and Appendix figures). Please also check that all the p-values are explained in the legend, and that these fit to those shown in the figure. Please provide statistical testing where applicable. Please avoid the phrase 'independent experiment', but clearly state if these were biological or technical replicates. Please also indicate (e.g. with n.s.) if testing was performed, but the differences are not significant. In case n=2, please show the data as separate datapoints without error bars and statistics. See also: <http://www.embopress.org/page/journal/14693178/authorguide#statisticalanalysis>

9) Please add scale bars of similar style and thickness to microscopic images, using clearly visible black or white bars (depending on the background). Please place these in the lower right corner of the images themselves. Please do not write on or near the bars in the image but define the size in the respective figure legend.

10) Please also note our reference format:

12) We now use CRedit to specify the contributions of each author in the journal submission system. CRedit replaces the author contribution section. Please use the free text box to provide more detailed descriptions and do NOT provide your final manuscript text file with an author contributions section. See also our guide to authors: <https://www.embopress.org/page/journal/14693178/authorguide#authorshipguidelines>

13) All Materials and Methods need to be described in the main text using our 'Structured Methods' format, which is required for

all research articles. According to this format, the Methods section should include a Reagents and Tools Table (listing key reagents, experimental models, software, and relevant equipment and including their sources and relevant identifiers), uploaded as separate file, and a Methods section in which we encourage the authors to describe their methods using a step-by-step protocol format with bullet points, to facilitate the adoption of the methodologies across labs. More information on how to adhere to this format as well as downloadable templates (.doc) for the Reagents and Tools Table can be found in our author guidelines (section 'Structured Methods'):

14) Please order the sections like this, using these names:

Title page - Abstract - Keywords - Introduction - Results & Discussion - Methods - Data availability section - Acknowledgements (including funding information) - Disclosure and Competing Interests Statement - References - Figure legends - Expanded View Figure legends

15) Please make sure that all the funding information is also entered into the online submission system and that it is complete and similar to the one in the acknowledgement section of the manuscript text file.

I look forward to seeing a revised form of your manuscript when it is ready.

Yours sincerely,

Referee #1:

Lim et al. investigated the role of CYLD in *H. pylori* ADP-heptose-induced activation of the NF- κ B pathway in gastric epithelial cells. The authors report that CYLD directly interacts with TRAF6 to upregulate NF- κ B responses and does so by mediating the deubiquitinylation of A20 in a catalytically independent manner. Furthermore, they show that NF- κ B-dependent upregulation of A20 provides a negative feedback loop on both classical and alternative NF- κ B signaling. The work provides new insights into *H. pylori* ADP-heptose-induced activation of the NF- κ B pathway in gastric epithelial cells. This transcription factor is critical to bacterial-induced inflammation and, therefore, the findings are likely to be of interest to not only workers in the *H. pylori* field, but also those working on other infections in which NF- κ B signaling is induced in response to ADP-heptose.

The manuscript is well written, and the data are nicely presented and clear. The authors have used two *H. pylori* strains and multiple knockout cell lines/models/siRNAs, thereby adding weight to their findings. The conclusions are generally supported by the data, though some points require further clarification or discussion. Specific points for the authors' consideration are listed below.

- 1) pg 4 - The authors describe the role of CYLD on activation of the classical NF- κ B pathway by *H. pylori* (lines 121-123). All of the evidence, however, is in the form of downstream proteins detected by Western blotting. Do they have any evidence that the responses translate to downstream NF- κ B-dependent responses, such as cytokine/chemokine production? i.e. Do the CYLD KO cells have reduced cytokine/chemokine production?
- 2) pg 4 - "ADP-heptose-triggered NF- κ B activation was attenuated in CYLD KO cells compared to WT cells (Fig. 1F)" (lines 132-133). The figure actually presents data for mouse spheroid cells, so should be referred to as *Cyld*^{-/-}, not "CYLD KO". It would also be important to confirm the role of ADP-heptose by stimulating CYLD WT and KO cells in the AGS cell line with this metabolite. And, or alternatively, they should infect these cells with *H. pylori* mutant bacteria that are unable to mediate ADP-heptose-induced activation of the NF- κ B pathway.

- 3) pg 4. The spheroids were generated from WT and *Cyld*^{-/-} mice originating from completely different sources and not litter-matched animals, or even animals from the same breeding colony (lines 307-311). Can the authors be sure that there are no confounding effects of the different strain backgrounds or gastric microbiomes on the results?
- 4) pg 4 - The authors discuss the double bands on CYLD blots as being due to phosphorylated and unphosphorylated forms of the protein (lines 113-116), however, there seem to be differences across the assays as to the presence of one or more of these bands. Could these differences be biologically significant? For example, in Fig. 2C, it seems that the phosphorylated form is absent when TIFA is also absent, whereas both forms were detected when TIFA was present. The authors need to discuss this in the manuscript.
- 5) pg 5 - I do not think that the overexpression data (lines 146-149, Fig. 1I) are particularly convincing. At the very least, the discussion regarding these data should be more nuanced.
- 6) Fig. 1 - The differences in IKK α /b and phospho-RelA levels between WT and KO cells do not seem that striking (Figs. 1A and F, respectively). Please comment. Fig. 1C would be clearer if the y-axis were in a log scale.
- 7) pg 5 - How can the authors conclude on the "presence of CYLD in the TIFA/TRAF6/TAK1 complex" (lines 174-175) when they also state that "TIFA was not required for the interaction between CYLD and TRAF6" (line 171). This can probably be addressed by a simple change in wording.
- 8) pg 6 - The effect of the siRNA KD of cIAP1 (line 212; Figs. 3D, E) seemed quite modest, so I do not think it is possible to be so definitive that there was "no effect on TRAF6 ubiquitinylation..." (line 214).
- 9) pg 7 - The meaning of the statement in lines 220-22 is unclear.
- 10) pg 9 - An MOI of 100 seems very high (lines 294-297). Do the authors see any differences between WT and CYLD KO cells when using an MOI of 10? Did they confirm actual numbers of viable bacteria? Otherwise, it is very difficult to assume an MOI from OD readings. How long were the bacteria incubated in contact with the cells?
- 11) pp 11-14 - The dilutions used for ALL antibodies must be reported.
- 12) It would be useful to have a schematic diagram describing the proposed pathway.
- 13) On several occasions, the authors write statements such as "We proceeded to use previously generated A20...cells" (line 222). If a reference is provided, there is no need to reiterate the point.

Referee #2:

The manuscript titled "CYLD-TRAF6 interaction promotes ADP-heptose-induced classical NF- κ B of *Helicobacter pylori*" highlights a novel role of CYLD in NF- κ B regulation. The authors found that CYLD interacts with TRAF6 in a complex with ALPK1/TIFA. CYLD triggers the ubiquitination of TRAF6 and thus the classical NF- κ B activity. The catalytic activity of CYLD is not required, which is in contrast to the known role of CYLD, which suppresses NF- κ B activity by removing K63- and M1 ubiquitin chains in cytokine-stimulated cells. The data and the current knowledge of CYLD in relation to NF- κ B were well evaluated and suggest a highly important role of CYLD in *Helicobacter* pathophysiology. Nevertheless, the reviewer has the following suggestions to improve the manuscript.

Specific points

- The introduction focuses on ADP-heptose-induced NF- κ B in *Helicobacter* infections. Are there other NF- κ B-inducing mechanisms in *Helicobacter* infection?
- To show more clearly whether the effect of CYLD on NF- κ B is ADP-Hep-dependent, the authors should use ADP-Hep as a trigger and ADP-Hep-deficient Hp strain in a TRAF6-IP (Fig. 2).
- The authors suggest that TRAF6 is auto-ubiquitinated and ruled out the role of cIAP1 by a knockdown experiment (Fig. 3D).
- If cIAP1 is not involved in TRAF6 ubiquitination, is cIAP2 expressed that could ubiquitinate TRAF6?
- To exclude the role of another E3 ligase, the ubiquitination of TRAF6 should be confirmed by transfection of a catalytically inactive TRAF6 mutant.
- It is assumed that TRAF6 is ubiquitinated with K63. Although this is to be expected, it should be confirmed in *Helicobacter* infections by using a K63-specific antibody.
- Regarding the TIFA/TRAF6/CYLD/A20 complex (Fig. 4A). Is there a physical binding between CYLD and A20?
- As mentioned above, ADP-Hep and ADP-Hep-deficient Hp strain should be used to show that the effects of CYLD on alternative NF- κ B are dependent on ADP-Hep (Fig. 5).
- Line 241: The translocation of RelB is mentioned, but is not shown in Fig. 5B.
- The authors show that CYLD promotes classical NF- κ B in *Helicobacter* infection and subsequently influences alternative NF- κ B via de novo synthesized A20 (Fig. 5). In the absence of de novo synthesized A20, alternative NF- κ B should not be affected by CYLD. Therefore, it would be interesting to show that stimulation of alternative NF- κ B, e.g. by lymphotoxin b, is not affected in CYLD knock-out cells.
- I would have liked to see an illustration of the specific signaling mechanism in *Helicobacter* infection in Fig. 5E. However, I know that the journal requires a graphical summary, but only if the manuscript is published.

Referee #3:

NF- κ B during *H. pylori* infection is crucial to the inflammatory response against *H. pylori*. In this manuscript, the authors

demonstrate that in contrast to CYLD's known inhibition of the NF- κ B pathway, during *H. pylori* infection, NF- κ B activation is reduced in a deubiquitinase independent manner. The authors demonstrate that CYLD interacts constitutively with TRAF6. CYLD expression also enhances the ubiquitination of TRAF6. This is also assumed to be independent of its enzymatic activity due to its effect on NF- κ B activation. An additional DUB enzyme, A20, decreases TRAF6 activation by its DUB activity. CYLD counteracts this effect by inhibiting the deubiquitination of TRAF6. Although the observation that CYLD enhanced the activation of NF- κ B is novel, the mechanism provided is largely speculative. Evidence of the necessity of CYLD and TRAF6 interaction and a regulatory motif of CYLD necessary for this effect would increase the impact of the manuscript.

Major concerns:

- There is no direct link between CYLD, TRAF6 and NF- κ B activation. All the pathways are speculative. One single point mutant that is not characterized in this manuscript and where no reference appeared to be available was the only evidence provided that the effect on NF- κ B activity was independent of the CYLD DUB activity. Attempts to provide a direct link through binding sites were all negative. A study of the motifs of CYLD necessary for interaction with TRAF6 were able to alter its effect on NF- κ B activity this would suggest a mechanism. Alternatively, interaction may be unnecessary for the effect of CYLD on NF- κ B.
- It's never shown that the enzymatic activity of CYLD is dispensable for ubiquitination of TRAF6. This should also be shown.
- While it's clear that A20 and CYLD have opposite effects on the ubiquitination of TRAF6, in the absence of a mechanism of CYLD, it's premature to speculate that the enhanced ubiquitination of TRAF6 is responsible for its effect on NF- κ B activation.

Minor Concerns:

- Line 166 "Reciprocal IP with a TIFA-directed antibody confirmed inducible association with TRAF6 and CYLD" seems to be counter to claims of a constitutive association and does not appear to be supported by the data. Please clarify.
- You claim that TRAF6 is auto-ubiquitinated, but multiple E3 enzymes may be involved. In the absence of a mechanism for the CYLD mediated enhancement of TRAF6 ubiquitination, this seems to be speculative.

Point-by-point response to the reviewers' reports

We thank the reviewers for their valuable comments and suggestions. Where appropriate, we have addressed the comments through additional experiments and have revised the figures accordingly. We have also revised the manuscript accordingly and made some minor changes to improve clarity.

Referee #1:

1) pg4 - The authors describe the role of CYLD on activation of the classical NF- κ B pathway by *H. pylori* (lines 121-123). All of the evidence, however, is in the form of downstream proteins detected by Western blotting. Do they have any evidence that the responses translate to downstream NF- κ B-dependent responses, such as cytokine/chemokine production? i.e. Do the CYLD KO cells have reduced cytokine/chemokine production?

We have examined the induction of NF- κ B-regulated transcripts, *TNF* and *IL-8*, to ascertain that the induction of the NF- κ B pathway indeed leads to a transcriptional response (Fig. 1C). In CYLD^{KO} cells, we observed a four-fold and ten-fold reduction of *TNF* and *IL-8* transcripts, respectively.

2) pg 4 - "ADP-heptose-triggered NF- κ B activation was attenuated in CYLD KO cells compared to WT cells (Fig. 1F)" (lines 132-133). The figure actually presents data for mouse spheroid cells, so should be referred to as *Cyld*^{-/-}, not "CYLD KO". It would also be important to confirm the role of ADP-heptose by stimulating CYLD WT and KO cells in the AGS cell line with this metabolite. And, or alternatively, they should infect these cells with *H. pylori* mutant bacteria that are unable to mediate ADP-heptose-induced activation of the NF- κ B pathway.

We regret that there was a misunderstanding regarding Fig. 1F. We have studied ADP-heptose-triggered classical NF- κ B activation in WT and CYLD^{KO} cells and the data are shown in Fig. 1F. In addition, we have provided a new supplementary figure (Fig. EV4A) showing additional data for ADP-heptose-triggered alternative NF- κ B activation in these cells.

We have included in our revised manuscript the following text (lines 262-263): 'The accumulation of NIK was also observed in ADP-heptose-treated CYLD^{KO} cells (Fig. EV4A).'

3) pg 4. The spheroids were generated from WT and *Cyld*^{-/-} mice originating from completely different sources and not litter-matched animals, or even animals from the same breeding colony (lines 307-311). Can the authors be sure that there are no confounding effects of the different strain backgrounds or gastric microbiomes on the results?

Both WT and *Cyld*^{-/-} mice share the same C57BL/6 background. The *Cyld*^{-/-} mice were backcrossed into C57BL/6 mice and therefore we consider their backgrounds as quite similar. However, since we used isolated gastric crypts from WT and *Cyld*^{-/-} mice to generate the spheroids, the genetic background should not affect the results. Furthermore, the microbiome of each mouse is unique, regardless of whether they are from the same or different litters.

To support our point, we have performed experiments with spheroids generated from isolated gastric crypts of different WT and *Cyld*^{-/-} mice and comparable results were yielded, as shown in the new supplementary figure (Fig. EV11).

4) pg 4 - The authors discuss the double bands on CYLD blots as being due to phosphorylated and unphosphorylated forms of the protein (lines 113-116), however, there seem to be differences across the assays as to the presence of one or more of these bands. Could these differences be biologically significant? For example, in Fig. 2C, it seems that the phosphorylated form is absent when TIFA is also absent, whereas both forms were detected when TIFA was present. The authors need to discuss this in the manuscript.

The absence of CYLD phosphorylation in Fig. 2C confirms that the CYLD phosphorylation is NF- κ B-dependent, as described by Elliott *et al.* (Elliott *et al.*, 2021). In the absence of TIFA, there is no binding of TAK1 to TRAF6 (Fig. 2C) and therefore there is no NF- κ B activation.

For clarification, we have revised the manuscript (lines 176-178): 'We also observed that CYLD phosphorylation was absent in TIFA^{KO} cells (Fig. 2C). According to Elliott and colleagues (Elliott *et al.*, 2021), CYLD phosphorylation is dependent on NF- κ B activation, which is abolished in TIFA^{KO} cells due to the missing interaction of TIFA with TRAF6 and TAK1.'

5) pg 5 - I do not think that the overexpression data (lines 146-149, Fig. 1I) are particularly convincing. At the very least, the discussion regarding these data should be more nuanced.

We have rephrased the discussion in a more nuanced manner as follows (lines 147-155): 'Overexpression of WT CYLD as well as catalytically inactive CYLD (C601A) in CYLD^{KO} cells enhanced classical NF- κ B signaling by *H. pylori* infection (Fig. 1I), indicating that CYLD's role in this pathway is not strictly dependent on its deubiquitinylase activity. While this does not entirely preclude a role for CYLD in modulating M1- and K63-linked ubiquitin chains during *H. pylori* infection, such activity appears to be of secondary relevance compared to its function in regulating receptor-mediated NF- κ B activation. These findings suggest that CYLD may influence classical NF- κ B signaling through mechanisms beyond direct ubiquitin editing, potentially implicating scaffolding or regulatory interactions that remain to be fully elucidated.'

6) Fig. 1 - The differences in IKK α /b and phospho-RelA levels between WT and KO cells do not seem that striking (Figs. 1A and F, respectively). Please comment. Fig. 1C would be clearer if the y-axis were in a log scale.

We do not expect any differences in the expression level of total IKK α /b (Fig. 1A). While the differences in the signal for phospho-RelA (Fig. 1F) are not that striking, they remain observable and significant.

The y-axis in Fig. 1C has been changed to log scale.

7) pg 5 - How can the authors conclude on the "presence of CYLD in the TIFA/TRAF6/TAK1 complex" (lines 174-175) when they also state that "TIFA was not required for the interaction between CYLD and TRAF6" (line 171). This can probably be addressed by a simple change in wording.

Our data show that CYLD interacts constitutively with TRAF6. Upon *H. pylori* infection, we observed the inducible interaction of TIFA with a complex comprising TRAF6/CYLD, which is necessary for the interaction between TAK1 and TRAF6 (Fig. 2C). Therefore, CYLD is present in the TIFA/TRAF6/TAK1 complex but does not require TIFA for interacting with TRAF6.

We have revised the manuscript (lines 179-180): 'Collectively, our results strongly indicated the constitutive interaction of CYLD with TRAF6 in the TIFA/TRAF6/TAK1 complex in *H. pylori* infection.'

8) pg 6 - The effect of the siRNA KD of cIAP1 (line 212; Figs. 3D, E) seemed quite modest, so I do not think it is possible to be so definitive that there was "no effect on TRAF6 ubiquitinylation..." (line 214).

We have revised the manuscript (lines 230-233): 'In addition, knockdown of cIAP1, another E3 ligase frequently involved in stimuli-induced NF- κ B activation (Mahoney *et al.*, 2008; Snelling *et al.*, 2022), had virtually no effect on TRAF6 ubiquitinylation and NF- κ B activation in response to *H. pylori* infection (Figs. 3E and 3F).'

9) pg 7 - The meaning of the statement in lines 220-22 is unclear.

We have revised the text (lines 240-243): 'This mechanism is plausible because in *H. pylori* infection, we detected the interaction of A20 with a complex composed of TRAF6, CYLD, TIFA and TAK1 (Fig. 4A), likely via A20's association with TIFA (Lim *et al.*, 2022) because recombinant A20 and CYLD did not interact directly (Fig. EV4E).'

10) pg 9 - An MOI of 100 seems very high (lines 294-297). Do the authors see any differences between WT and CYLD KO cells when using an MOI of 10? Did they confirm actual numbers of viable bacteria? Otherwise, it is very difficult to assume an MOI from OD readings. How long were the bacteria incubated in contact with the cells?

We observed under the microscope that the bacteria were highly motile after addition to the cells, indicating their viability. Using an OD₅₅₀ measurement of *H. pylori* to estimate the MOI is common practice in this research field. The MOI estimation is done using a growth curve or the formula $y=2 \times 10^8 * \ln(OD_{550}) + 5 \times 10^8$. The bacteria were incubated with the cells for the durations specified in the respective experiments.

We also observed differences between WT and CYLD^{KO} cells when we used an MOI of 10 for infection (see below).

11) pp 11-14 - The dilutions used for ALL antibodies must be reported.

We have included the dilutions for all antibodies in the 'Reagents and Tools Table'.

12) It would be useful to have a schematic diagram describing the proposed pathway.

We have included a schematic diagram describing the proposed pathway as Fig. 5E.

13) On several occasions, the authors write statements such as "We proceeded to use previously generated A20...cells" (line 222). If a reference is provided, there is no need to reiterate the point.

We have deleted the unnecessary reiterations.

Referee #2:

Specific points

- The introduction focuses on ADP-heptose-induced NF-κB in Helicobacter infections. Are there other NF-κB-inducing mechanisms in Helicobacter infection?

Yes, there are other mechanisms described in the literature on how *H. pylori* induces NF-κB (Backert & Naumann, 2010). We have included the mention of this in the manuscript (lines 42-43): 'Multiple bacterial factors have been implicated in the induction of NF-κB upon *H. pylori* infection (Backert & Naumann, 2010).'

-To show more clearly whether the effect of CYLD on NF-κB is ADP-Hep-dependent, the authors should use ADP-Hep as a trigger and ADP-Hep-deficient Hp strain in a TRAF6-IP (Fig. 2).

ADP-heptose was used as a trigger for classical NF-κB activation and a TRAF6 IP was performed (Fig. EV2B). We have revised the manuscript (lines 169-170): 'Similar effects were observed using ADP-heptose as trigger (Fig. EV2B).'

The ADP-heptose-deficient strain would not induce NF-κB and therefore a TRAF6-associated complex is not expected.

- The authors suggest that TRAF6 is auto-ubiquitinated and ruled out the role of cIAP1 by a knockdown experiment (Fig. 3D).
- If cIAP1 is not involved in TRAF6 ubiquitination, is cIAP2 expressed that could ubiquitinate TRAF6?

The cIAP2 upregulation was observed after 90 min of *H. pylori* infection (new Fig. EV3D). We observed a difference in TRAF6 ubiquitinylation between WT and CYLD^{KO} cells already within 30 min of *H. pylori* infection (Fig. 3A). Therefore, it is unlikely that cIAP2 plays a role in TRAF6 ubiquitinylation.

We have included this data and revised the manuscript (lines 233-235): 'The functionally redundant cIAP2 is upregulated only after 90 minutes of *H. pylori* infection (Fig. EV3D) and is therefore unlikely to play a role in the TRAF6 ubiquitinylation that we observed as early as 30 min post infection.'

- To exclude the role of another E3 ligase, the ubiquitination of TRAF6 should be confirmed by transfection of a catalytically inactive TRAF6 mutant.

We have performed the overexpression of a catalytically inactive TRAF6 mutant and observed a reduction in TRAF6 ubiquitinylation (new Fig. 3D). Furthermore, we have corroborated this result by using a TRAF6-Ubc13 selective inhibitor that abolishes TRAF6's E3 ligase activity (new Fig. EV3C).

We have revised the manuscript (lines 227-230): 'The TRAF6 ubiquitinylation observed in *H. pylori* infection is most likely auto-ubiquitinylation because overexpression of a dominant-negative catalytically inactive TRAF6 mutant (Fig. 3D) as well as pre-incubation of the cells with C25-140 (Fig. EV3C) significantly reduced TRAF6 ubiquitinylation.'

- It is assumed that TRAF6 is ubiquitinated with K63. Although this is to be expected, it should be confirmed in *Helicobacter* infections by using a K63-specific antibody.

We currently do not have a commercially available K63-linked ubiquitin-specific antibody that performs effectively in our experiments. Our experiment involving the catalytically inactive TRAF6 mutant established the presence of TRAF6 auto-ubiquitinylation (Fig. 3D). As the activity of TRAF6 is K63-linked ubiquitin-specific (Lamothe *et al*, 2007), this experiment confirms the K63-linked ubiquitinylation of TRAF6.

- Regarding the TIFA/TRAF6/CYLD/A20 complex (Fig. 4A). Is there a physical binding between CYLD and A20?

We have examined the direct physical binding of A20 and CYLD by co-incubating recombinant human A20 and CYLD proteins and performing an A20 IP. Compared to the positive control that is the interaction between A20 and TIFA, no significant interaction between A20 and CYLD was detected.

We have included this data and revised the manuscript accordingly (lines 240-243): 'This mechanism is plausible because in *H. pylori* infection, we detected the interaction of A20 with a complex composed of TRAF6, CYLD, TIFA and TAK1 (Fig. 4A), likely via A20's association with TIFA (Lim *et al*, 2022) because recombinant A20 and CYLD did not interact directly (Fig. EV3E).'

- As mentioned above, ADP-Hep and ADP-Hep-deficient Hp strain should be used to show that the effects of CYLD on alternative NF- κ B are dependent on ADP-Hep (Fig. 5).

We have performed the experiment with ADP-heptose and have shown the results in the new supplementary figure (Fig. EV4A). We have revised the manuscript according (lines 262-263): 'The accumulation of NIK was also observed in ADP-heptose-treated CYLD^{KO} cells (Fig. EV4A).' The ADP-heptose-deficient strain does not induce alternative NF- κ B (Maubach *et al*, 2021).

- Line 241: The translocation of RelB is mentioned, but is not shown in Fig. 5B.

We have amended the sentence by removing the mention of RelB.

- The authors show that CYLD promotes classical NF- κ B in Helicobacter infection and subsequently influences alternative NF- κ B via de novo synthesized A20 (Fig. 5). In the absence of de novo synthesized A20, alternative NF- κ B should not be affected by CYLD. Therefore, it would be interesting to show that stimulation of alternative NF- κ B, e.g. by lymphotoxin b, is not affected in CYLD knock-out cells.

We have stimulated WT and CYLD^{KO} cells with LT $\alpha_1\beta_2$ to induce the alternative NF- κ B pathway. We observed that since there was no *de novo* synthesis of A20 in these cells, there was also no difference in the activation of the alternative NF- κ B pathway between them (new Fig. EV4B). We have revised the manuscript accordingly (lines 263-264): 'In LT $\alpha_1\beta_2$ -treated CYLD^{KO} cells, however, there was no accumulation of NIK since the abundance of A20 remained the same (Fig. EV4B).'

- I would have liked to see an illustration of the specific signaling mechanism in Helicobacter infection in Fig. 5E. However, I know that the journal requires a graphical summary, but only if the manuscript is published.

We have included a schematic diagram describing the proposed pathway as new Fig. 5E.

Referee #3:

Major concerns:

•There is no direct link between CYLD, TRAF6 and NF- κ B activation. All the pathways are speculative. One single point mutant that is not characterized in this manuscript and where no reference appeared to be available was the only evidence provided that the effect on NF- κ B activity was independent of the CYLD DUB activity. Attempts to provide a direct link through binding sites were all negative. A study of the motifs of CYLD necessary for interaction with TRAF6 were able to alter its effect on NF- κ B activity this would suggest a mechanism. Alternatively, interaction may be unnecessary for the effect of CYLD on NF- κ B.

The catalytically inactive CYLD mutant referred to in Fig. 1I was previously published (Stegmeier *et al*, 2007) and is referenced in the 'Materials and methods' section. As suggested by the reviewer, we have studied the motifs of CYLD that might be necessary for its interaction with TRAF6 by overexpressing HA-tagged full length as well as truncated variants of CYLD and FLAG-tagged TRAF6. These CYLD variants include the three CAP-Gly domains, a middle domain and the C-terminal USP domain. By performing a FLAG or HA IP, we precipitated the full length as well as truncated variants of CYLD with the exception of the variant containing only the first CAP-Gly domain (1-220) (data are appended as a new figure, Fig. 2I). This finding suggested that the N-terminal portion of CYLD is not required for its interaction with TRAF6.

We have included this data in the manuscript (lines 201-207): 'To further elucidate the region of CYLD that is implicated in the interaction with TRAF6, we transfected plasmids expressing HA-tagged full length and truncated variants of human CYLD into CYLD^{KO} cells together with FLAG-tagged TRAF6 (Figs. 2H and 2I) and analyzed their binding ability to TRAF6 by performing a FLAG or HA IP. We precipitated the full length as well as truncated variants of CYLD (Fig. 2I) with the exception of the variant containing only the first CAP-Gly domain (1-220). This finding suggested that the N-terminal portion of CYLD is not required for its interaction with TRAF6.'

•It's never shown that the enzymatic activity of CYLD is dispensable for ubiquitination of TRAF6. This should also be shown.

We have now shown that the catalytically inactive CYLD mutant (pCYLD C601A) has no effect on TRAF6 ubiquitinylation (Fig. EV3B).

•While it's clear that A20 and CYLD have opposite effects on the ubiquitination of TRAF6, in the absence of a mechanism of CYLD, it's premature to speculate that the enhanced ubiquitination of TRAF6 is responsible for its effect on NF- κ B activation.

We have now shown that incubation of the cells with a selective TRAF6-Ubc13 inhibitor, C25-140, (Brenke *et al*, 2018) prior to *H. pylori* infection resulted in attenuated classical NF- κ B activation (reduced phosphorylation of TAK1 and IKK α/β) and TRAF6 ubiquitinylation (data are appended as new Figs. EV3A and EV3C). Our results implicate TRAF6 ubiquitinylation in *H. pylori*-induced classical NF- κ B activation.

We have revised the manuscript accordingly (lines 216-219): 'Incubation of the cells with a selective TRAF6-Ubc13 inhibitor, C25-140, (Brenke *et al*, 2018) prior to *H. pylori* infection led to the reduced phosphorylation of TAK1 and IKK α/β (Fig. EV4A), implicating TRAF6 auto-ubiquitinylation in *H. pylori*-induced classical NF- κ B activation.'

Minor Concerns:

•Line 166 "Reciprocal IP with a TIFA-directed antibody confirmed inducible association with TRAF6 and CYLD" seems to be counter to claims of a constitutive association and does not appear to be supported by the data. Please clarify.

We did not see the constitutive interaction of TIFA with TRAF6 and CYLD (Fig. EV2C). Rather, we observed that upon *H. pylori* infection, TIFA interacts with TRAF6, which already has bound CYLD (Fig. EV2C), which is consistent with our data showing constitutive interaction between TRAF6 and CYLD (Fig. 2A).

•You claim that TRAF6 is auto-ubiquitinated, but multiple E3 enzymes may be involved. In the absence of a mechanism for the CYLD mediated enhancement of TRAF6 ubiquitination, this seems to be speculative.

We have now shown that in *H. pylori* infection, overexpression of a catalytically inactive TRAF6 mutant had a dominant-negative effect on TRAF6 ubiquitinylation (Fig. 3D) and pre-incubation of the cells with a selective TRAF6-Ubc13 inhibitor, C25-140, also reduced TRAF6 ubiquitinylation (Fig. EV3C). These findings suggest that TRAF6 is auto-ubiquitinated.

Dear Prof. Naumann,

Thank you for the submission of your revised manuscript to our editorial offices. I have now received the reports from the three referees that were asked to re-evaluate the study, you will find below. As you will see, the referees now support its publication in EMBO reports. Referee #3 has a few remaining concerns and suggestions to improve the manuscript, I ask you to address in a final revised manuscript. Please also provide a final p-b-p-response regarding these points and the editorial requests below.

I have these editorial requests:

- I suggest this revised title:

CYLD-TRAF6 interaction promotes ADP-heptose-induced NF- κ B signaling in *H. pylori*

- Please order the manuscript sections like this, using these names:

Title page - Abstract - Keywords - Introduction - Results & Discussion - Methods - Data availability section - Acknowledgements

- Disclosure and Competing Interests Statement - References - Figure legends - Expanded View Figure legends

- We now use CRediT to specify the contributions of each author in the journal submission system. CRediT replaces the author contribution section. Please use the free text box to provide more detailed descriptions and do NOT provide your final manuscript text file with an author contributions section. See also our guide to authors:

<https://www.embopress.org/page/journal/14693178/authorguide#authorshippinguidelines>

- There is an author name discrepancy. It is Michelle C.C. Lim in the manuscript text file but Michelle Lim in the submission system. Please check.

- Please confirm that for all Western blot panels (main and EV figures) the loading control was run on the same gel as the other proteins detected. Please note that we discourage comparisons between samples on different gels/blots, even if the samples derive from one experiment, as confounding factors reduce comparability. If unavoidable, the figure legend must state that the samples derive from the same experiment and that gels/blots were processed in parallel. If a 'representative' loading control is shown for multiple gels/blots, the intra-gel controls should be shown in the source data files and the figure legends should describe the data displayed accurately. See our author guidelines:

<https://www.embopress.org/page/journal/14693178/authorguide#datapresentationformat> (section 'Electrophoretic gels and blots').

and

<https://www.embopress.org/image-integrity>

- Please check again that the number "n" for how many independent experiments were performed, their nature (biological versus technical replicates), the bars and error bars (e.g. SEM, SD) and the test used to calculate p-values is indicated in the respective figure legends. Please also check that all the p-values are explained in the legend, and that these fit to those shown in the figure. Please provide statistical testing where applicable. Please avoid the phrase 'independent experiment', but clearly state if these were biological or technical replicates. Please also indicate (e.g. with n.s.) if testing was performed, but the differences are not significant. In case n=2, please show the data as separate datapoints without error bars and statistics. See also:

<http://www.embopress.org/page/journal/14693178/authorguide#statisticalanalysis>

If n<5, please show single datapoints for diagrams. Moreover:

- Please provide the exact p values are in the legend of figure 1C.

- Please add to each legend (main and EV figures, where applicable) a 'Data Information' section (or name the provided section like this) explaining the statistics used or providing information regarding replicates and scales. See:

- Please provide the Reagents and Tools Table in .doc format. Please find the downloadable template (.doc) for the Reagents and Tools Table in our author guidelines (section 'Structured Methods'):

- Please move the primer information (uploaded as Table S1) to the 'Reagents & Tools Table' and update any callouts.

In addition, I would need from you uploaded separately:

Best,

Referee #1:

The authors have satisfactorily addressed to the reviewers' comments.

Referee #2:

All my questions have been answered adequately, I recommend publication in EMBO Reports.

Referee #3:

Thanks for all the thorough responses to all reviewer comments.

The TRAF6 mutant and inhibitor data strongly suggest that auto-ubiquitination is responsible.

I still have a few minor issues.

I did not see if you were able to test if the CYLD KO cells could still alter NF-KB with the TRAF6 inhibitor or the TRAF6 mutant? If not, thus would suggest that CYLD is working through the TRAF6 pathway. Please let me know if I missed these experiments.

The methods did not mention where the catalytically inactive mutant was acquired, and it was therefore difficult to determine whether it had been characterized. The reference to the wild-type plasmid "HA-CYLD-WT" should be updated to include the mutant.

Point-by-point response to editorial requests and reviewer comments

Editorial requests

- I suggest this revised title:
CYLD-TRAF6 interaction promotes ADP-heptose-induced NF- κ B signaling in *H. pylori*

We have revised the title as follows 'CYLD-TRAF6 interaction promotes ADP-heptose-induced NF- κ B signaling in *H. pylori* infection'

- Please order the manuscript sections like this, using these names:
Title page - Abstract - Keywords - Introduction - Results & Discussion - Methods - Data availability section - Acknowledgements - Disclosure and Competing Interests Statement - References - Figure legends - Expanded View Figure legends

We have ordered the manuscript sections as indicated above.

- We now use CRediT to specify the contributions of each author in the journal submission system. CRediT replaces the author contribution section. Please use the free text box to provide more detailed descriptions and do NOT provide your final manuscript text file with an author contributions section. See also our guide to authors: <https://www.embopress.org/page/journal/14693178/authorguide#authorshippinguidelines>

We have deleted the 'Author contributions' section from the manuscript and have used CRediT to specify the contributions of each author.

- There is an author name discrepancy. It is Michelle C.C. Lim in the manuscript text file but Michelle Lim in the submission system. Please check.

The author name is Michelle C.C. Lim, which was not possible to correct in the submission system.

- Please confirm that for all Western blot panels (main and EV figures) the loading control was run on the same gel as the other proteins detected. Please note that we discourage comparisons between samples on different gels/blots, even if the samples derive from one experiment, as confounding factors reduce comparability. If unavoidable, the figure legend must state that the samples derive from the same experiment and that gels/blots were processed in parallel. If a 'representative' loading control is shown for multiple gels/blots, the intra-gel controls should be shown in the source data files and the figure legends should describe the data displayed accurately. See our author guidelines: <https://www.embopress.org/page/journal/14693178/authorguide#datapresentationformat> (section 'Electrophoretic gels and blots') and <https://www.embopress.org/image-integrity>

We used gels of varying percentages for detecting different proteins, depending on their molecular weights. Therefore, it is not possible to detect all the proteins, including the loading control for a given panel, on the same gel. However, we can confirm that for each Western blot panel, samples are derived from the same experiment and the gels/blots were processed in parallel. We have revised the figure legends accordingly in the figure legends '*immunoblots (IBs) were processed in parallel and depict 1 representative of at least two independent experiments.*'

- Please check again that the number "n" for how many independent experiments were performed, their nature (biological versus technical replicates), the bars and error bars (e.g. SEM, SD) and the test used to calculate p-values is indicated in the respective figure legends. Please also check that all the p-values are explained in the legend, and that these fit to those shown in the figure. Please provide statistical testing where applicable. Please avoid the phrase 'independent experiment', but clearly state if these were biological or technical replicates. Please also indicate (e.g. with n.s.) if testing was performed, but the differences are not significant. In case n=2, please show the data as separate datapoints without error bars and statistics. See also: <http://www.embopress.org/page/journal/14693178/authorguide#statisticalanalysis>. If n<5, please show single datapoints for diagrams. Moreover:- Please provide the exact p values are in the legend of figure 1C.

We have revised Fig. 1C accordingly. Since n = 3, we have amended the diagram to show single datapoints.

- Please add to each legend (main and EV figures, where applicable) a 'Data Information' section (or name the provided section like this) explaining the statistics used or providing information regarding replicates and scales. See: <https://www.embopress.org/page/journal/14693178/authorguide#figureformat>

We have renamed the provided section as 'Data information' (line 497).

- Please provide the Reagents and Tools Table in .doc format. Please find the downloadable template (.doc) for the Reagents and Tools Table in our author guidelines (section 'Structured Methods'): <https://www.embopress.org/page/journal/14693178/authorguide#manuscriptpreparation>

We have provided the Reagents and Tools Table in .doc format.

- Please move the primer information (uploaded as Table S1) to the 'Reagents & Tools Table' and update any callouts.

We have updated and moved the primer information to the 'Reagents and Tools Table'.

In addition, I would need from you uploaded separately:

a short, two-sentence summary of the manuscript (not more than 35 words).

two to four short (!) bullet points highlighting the key findings of your study (two lines each).

a schematic summary figure as separate file that provides a sketch of the major findings (not a data image) in jpeg or tiff format (with the exact width of 550 pixels and a height of not more than 400 pixels) that can be used as a visual synopsis on our website.

We have uploaded separately the summary, key findings and a schematic summary figure.

Reviewer #3

I did not see if you were able to test if the CYLD KO cells could still alter NF- κ B with the TRAF6 inhibitor or the TRAF6 mutant? If not, this would suggest that CYLD is working through the TRAF6 pathway. Please let me know if I missed these experiments.

Classical NF- κ B activation in *H. pylori* infection is controlled by the ALPK1-TIFA-TRAF6 axis. To demonstrate the importance of TRAF6 in classical NF- κ B signaling, we used a selective TRAF6-Ubc13 inhibitor that resulted in attenuated classical NF- κ B activation (Fig. EV3A). Furthermore, we showed that ADP-heptose treatment in CYLD KO cells resulted in the same outcome (reduced classical NF- κ B activation) as in *H. pylori* infection (Fig. 1A and 1F). Further, our additional data of CYLD/TRAF6 interaction indicate that the effect of CYLD abrogation on the NF- κ B signaling pathway is TRAF6-dependent. Therefore, we refrain from testing whether the attenuation of NF- κ B signaling still occurs in CYLD KO cells when we use the TRAF6 inhibitor.

The methods did not mention where the catalytically inactive mutant was acquired, and it was therefore difficult to determine whether it had been characterized. The reference to the wild-type plasmid "HA-CYLD-WT" should be updated to include the mutant.

We had cited the origin of the mutants and the wt constructs in the previous document in the lines (364 and 402).

Prof. Michael Naumann
Otto von Guericke University
Institute of Experimental Internal Medicine, Medical Faculty
Leipziger Str. 44
Magdeburg
Germany

Dear Prof. Naumann,

I am very pleased to accept your manuscript for publication in the next available issue of EMBO reports. Thank you for your contribution to our journal.

Yours sincerely,
